# The evolution of biomass-burning aerosol size distributions due to coagulation: dependence on fire and meteorological details and parameterization

**K.M. Sakamoto[1], J.R. Laing[2], R.G. Stevens[3], D.A. Jaffe[2,4], and J. R. Pierce[1,5]**

[1] Colorado State University, Fort Collins, CO, USA

[2] University of Washington-Bothell, Bothell, WA, USA

[3] University of Leeds, Leeds, UK

[4] University of Washington, Seattle, WA, USA

[5] Dalhousie University, Halifax, NS, Canada

Corresponding author: J.R. Pierce (jeffrey.pierce@colostate.edu)

## Abstract

Biomass-burning aerosols have a significant effect on global and regional aerosol climate forcings. To model the magnitude of these effects accurately requires knowledge of the size distribution of the emitted and evolving aerosol particles. Current biomass-burning inventories do not include size distributions, and global and regional models generally assume a fixed size distribution from all biomass-burning emissions. However, biomass-burning size distributions evolve in the plume due to coagulation and net organic aerosol (OA) evaporation or formation, and the plume processes occur on spacial scales smaller than global/regional-model grid boxes. The extent of this size-distribution evolution is dependent on a variety of factors relating to the emission source and atmospheric conditions. Therefore, to account for biomass-burning aerosol size in global models accurately requires an *effective* aerosol size distribution that accounts for this sub-grid evolution and can be derived from available emissions-inventory and meteorological parameters.

In this paper, we perform a detailed investigation of the effects of coagulation on the aerosol size distribution in biomass-burning plumes. We compare the effect of coagulation to that of OA evaporation and formation. We develop coagulation-only parameterizations for effective biomass-

burning size distributions using the SAM-TOMAS large-eddy simulation plume model. For the most-
sophisticated parameterization, we use the Gaussian Emulation Machine for Sensitivity Analysis
(GEM-SA) to build a parameterization of the aged size distribution based on the SAM-TOMAS output
and seven inputs: emission median dry diameter, emission distribution modal width, mass emissions
flux, fire area, mean boundary-layer wind speed, plume mixing depth, and time/distance since
emission. This parameterization was tested against an independent set of SAM-TOMAS simulations,
and yields $R^2$ values of 0.83 and 0.89 for $D_{pm}$ and modal width, respectively. The size distribution is
particularly sensitive to the mass emissions flux, fire area, wind speed, and time, and we provide
simplified fits of the aged size distribution to just these input variables. The simplified fits were tested
against eleven aged biomass-burning size distributions observed at the Mt. Bachelor Observatory in
August 2015. The simple fits captured over half of the variability in observed $D_{pm}$ and modal width
even though the freshly emitted $D_{pm}$ and modal widths were unknown. These fits may be used in global
and regional aerosol models. Finally, we show that coagulation generally leads to greater changes in the
particle size distribution than does OA evaporation/formation using estimates of OA production/loss
from the literature.

# 1. Introduction


## *1.1 Biomass-burning aerosols*


Biomass burning (including wildfires, prescribed fires, and agricultural fires) releases significant
amounts of gas- and particle-phase species to the atmosphere (Andreae and Merlet, 2001; Reid et al.,
2005). The particle-phase emissions are composed primarily of a mixture of organic aerosol (OA) and
black carbon (BC) with some inorganic species (e.g. potassium), and the ratios of these species depend
on the source fire conditions (Capes et al., 2008; Carrico et al., 2010; Cubison et al., 2011; Hecobian et
al., 2011; Hennigan et al., 2011; Reid et al., 2005). These aerosols affect the global radiation budget
through the indirect and direct aerosol effects (Boucher et al., 2013). The smoke particles themselves
are able to act as cloud condensation nuclei (CCN) and increase cloud albedo and lifetime (indirect
aerosol effect; Lee et al., 2013; Pierce et al., 2007; Spracklen et al., 2011) as well as
scattering/absorbing incoming solar-radiation directly (direct aerosol effect; Alonso-Blanco et al., 2014;
Boucher et al., 2013; Haywood and Boucher, 2000; Jacobson, 2001).

Particle size has a significant effect on the magnitude of both the direct and indirect aerosol

effects (Lee et al., 2013; Seinfeld and Pandis, 2006; Spracklen et al., 2011). The composition and
diameter of the particles affect their absorption/scattering efficiencies, which dictate the amount of
solar radiation absorbed/scattered per emitted mass of particles (Seinfeld and Pandis, 2006). Particle
diameter and hygroscopicity determine the particles' ability to act as a CCN and influence cloud
processes, and the total number of emitted particles increases with decreased particle size if total mass
emissions are fixed. Spracklen et al., (2011) found that a reduction by a factor of two in particle size for
all carbonaceous aerosols (for a fixed total aerosol mass) resulted in a ~300% increase in the cloud
albedo indirect effect globally, as more particles were available to act as CCN. Lee et al., (2013)
determined that CCN concentrations in the GLOMAP model were very sensitive to uncertainties in
biomass-burning emission diameter on both the regional and global scale (its attributable CCN
uncertainty ranked third of 28 factors tested globally). Therefore, to ascertain the role of biomass-
burning aerosols in climate forcings accurately, biomass-burning size distributions must be well
represented in aerosol-climate models.

Size distributions are subject to physical and chemical processing in the plume. The formation of

secondary organic aerosol (SOA) has been observed in lab studies of biomass-burning aerosol
(Cubison et al., 2011; Grieshop et al., 2009; Hennigan et al., 2011; Heringa et al., 2011; Ortega et al.,
2013) and in field campaigns (DeCarlo et al., 2010; Lee et al., 2008; Reid et al., 1998; Yokelson et al.,
2009). This SOA can condense onto existing particles causing growth of the aerosol size distribution. It
can also spur new-particle formation in biomass-burning plumes as has been observed in lab studies
(Hennigan et al., 2012) and field campaign analyses (Vakkari et al., 2014). Conversely, recent lab and
field studies have characterized primary organic aerosol (POA) as semi-volatile, with plume dilution
allowing the evaporation of organic aerosol from particles (Huffman et al., 2009; Cubison et al., 2011;
May et al., 2013, 2015; Jolleys et al., 2015). The cumulative net effects of OA production/loss within
biomass-burning plumes has been found to be highly variable from fire to fire (Akagi et al., 2012;
Hennigan et. al, 2011).

Coagulation is also important for size-distribution evolution as it reduces particle number and

shifts the distribution to larger sizes. Coagulation rates are proportional to the square of the particle
number concentration (all else remaining fixed), so the high number concentrations in biomass-burning
plumes relative to background can lead to rapid coagulational growth of the size distribution. The
coagulation rate is therefore also affected by the rate of plume dilution (through a reduction in N), itself
a function of plume size and meteorological conditions. The rate and magnitude of the aerosol growth
caused by these combined processes is a function of aging time, emission source characteristics,
aerosol properties at emission, and atmospheric conditions.
These condensation/evaporation and coagulation aging processes affect both the composition and
size of the aerosol size distribution – both properties that influence the extent to which smoke particles
affect climate. While fresh smoke is generally composed of fine particles between 20-60 nm in
diameter (Levin et al., 2010), condensation and coagulation cause rapid aerosol growth to larger sizes
(over 100 nm) on timescales of often less than 24 hrs (Janhäll et al., 2010). However, Janhäll et al.,
(2010) found the observed geometric mean diameter of aged biomass-burning particles varied between
170-300 nm, with geometric standard deviations (hereafter referred to as "modal width") between 1.3-
1.7 with significant dependence on fuel type and modified combustion efficiency. It is currently unclear
to what extent these factors and others drive the variability in aged size distributions.
As stated earlier, an accurate representation of aged biomass-burning aerosol size is necessary for
predictions of aerosol climate effects in regional and global models (Lee et al., 2013). Current wildfire
inventories are mass-based (neglecting aerosol size data), and thus regional and global models used for
aerosol-climate effects generally specify fixed, "aged" size distributions that do not account for sub-
grid processing of the emitted particles (Reid et al., 2009; van der Werf et al., 2010; Wiedinmyer et al.,
2011). Any variability in the biomass-burning size distribution due to fire or emissions characteristics
and meteorology are not accounted for, nor is it clear what the best "aged" size distribution to use is in
these models.
In this paper, we perform a detailed investigation of coagulation in biomass-burning plumes and
compare to the effects of OA evaporation and formation. We investigate the factors that influence
coagulational growth of the particles in the plume. These factors include fire area, particle-emissions
mass flux, particle-emissions size, and meteorological conditions. We create parameterizations of
varying degrees of complexity for median dry diameter ($D_{pm}$) and lognormal modal width ($\sigma$) of the
aged biomass-burning size distributions as a function of these input parameters, based on detailed
numerical simulations using a large-eddy model with embedded aerosol microphysics (SAM-TOMAS).
Finally, we compare the effect of coagulation on the aerosol size distribution to that of OA
production/evaporation.
We describe the parameterization building process, including the use of a Gaussian emulator, in
Sect. 2. A discussion of input and output ranges, processing, and constraints of the parameters we have
chosen is provided in Sect. 2.1. We discuss the SAM-TOMAS model and the emulation process in Sect.
2.2-2.3. Sections 3.1-3.2 contain the results of the SAM-TOMAS model and the emulator. We discuss
emulator sensitivities to the inputs in Sect. 3.3 and present a series of simplified fit equations for the
effective size distributions in Sect. 3.4. We discuss the effects of potential OA production/loss on our
size distribution estimates in Sect. 3.5. The simplified-fit equations are tested against biomass-burning
plumes observed at the Mt. Bachelor Observatory in Sect. 3.6. Finally, we conclude in Sect. 4,
including future plans for testing the parameterization and known existing limitations.

## 2. Methods

Figure 1 provides an overview of our methods that will be described in detail in the subsections below.
In short, we used a Large-Eddy Simulation model, the System for Atmospheric Modelling (SAM;
Khairoutdinov and Randall, 2003), with the online aerosol microphysics module, TwO Moment
Aerosol Sectional (TOMAS, Adams and Seinfeld, 2002; Stevens et al., 2012) to simulate the evolution
of the biomass-burning aerosol size distribution by coagulation across a wide range of emission and
meteorological conditions. We used the SAM-TOMAS size distributions to build parameterizations to
predict aged $D_{pm}$ and $\sigma$ using: (1) a statistical emulator of the SAM-TOMAS model itself and (2)
simplified fits to the SAM-TOMAS output data. The statistical emulator was built by the Gaussian
Emulation Machine for Sensitivity Analysis (GEM-SA), and we used the emulator and SAM-TOMAS
data to determine the relative importance of various inputs to shaping the aged size distribution.

### *2.1 Investigated factors that may lead to variability in aged size distributions*

We investigated seven parameters that may affect the aging of the biomass-burning aerosol size
distribution. These can be divided into those representing the initial lognormal-mode size parameters
($D_{pm0}$, $\sigma_0$), fire conditions (mass flux, fire area), atmospheric conditions (wind speed, plume mixing
depth), and time. Each of these parameters is generally available in large-scale aerosol models, which
means a parameterization for aged biomass-burning size distributions based on these parameters may
be used in these models. Table 1 lists these input parameters and the ranges of values tested in this
work.

We assumed that the initial size distributions were a single lognormal mode (described by dry

median diameter, $D_{pm}$, and modal width, $\sigma$), which is sufficient when representing both fresh and aged
observed biomass-burning size distributions (Capes et al., 2008; Janhäll et al., 2010; Levin et al., 2010;
Sakamoto et al., 2015). The initial size-distribution parameters specify the median dry diameter ($D_{pm0}$)
and modal width ($\sigma_0$) of the freshly emitted aerosol distribution. We varied these parameters between
20-100 nm for $D_{pm0}$ and 1.2-2.4 for $\sigma_0$. The large ranges are due to variability in combustion efficiency
and fuel-type factors as seen in lab and observational studies (Janhäll et al., 2010; Levin et al., 2010).

Fire area, mass flux, wind speed and aerosol mixing depth (hereafter referred to as *mixing*

*depth;* the vertical extent of the aerosol plume) all affect the aerosol number concentration (N) within
the plume, which in turn affects the coagulation rate (proportional to $N^2$). In our simulations, we
constrained mass flux to $2 \times 10^{-8}$ - $5 \times 10^{-6}$ kg m$^{-2}$ s$^{-1}$ using approximate maximum and minimum values
of summed black carbon and organic carbon flux (BC+OC) found in the Global Fire Emissions
Database ver. 3 (GFED3; van der Werf et al., 2010; available from http://www.globalfiredata.org). Fire
area ranged from 1 - 49 km$^2$ (simulated as a square), which was found to represent the range of fire
sizes in GFED3. Boundary layer wind speed varied between 2 m s$^{-1}$ and 20 m s$^{-1}$ and was based on
ranges in the National Center for Environmental Prediction (NCEP) North American Regional
Reanalysis (NARR) meteorology (Mesinger et al., 2006) during the fire season (specifically, July,
2010). Mixing depth had a range of 150-2500 m (based on SAM-TOMAS output; see Sect. 2.2).

The aging time was the final input parameter, and we used 5 hr (300 min) as an upper time

bound due to this being a typical timescale for transport across large global model gridboxes.

### 2.2 The SAM-TOMAS model


We used the SAM-TOMAS model to simulate the evolution of biomass-burning aerosol size
distributions due to coagulation across the range of input parameters described above. SAM
(Khairoutdinov and Randall, 2003) is a dynamical large-eddy simulation (LES) model, which has
previously been used to model emissions plumes (Lonsdale et al., 2012; Stevens et al., 2012; Stevens
and Pierce, 2013). We ran the model in Lagrangian 2D mode (Stevens and Pierce, 2013), in which a
wall oriented normal to the mean boundary layer wind moves at the mean boundary-layer wind speed.
This moving wall tracks the radial dispersion of a plume as it travels downwind (Fig. 2). This 2D mode
is computationally efficient compared to the full 3D model with minor differences due to axial plume
symmetry (Stevens and Pierce, 2013).

The size distributions of the aerosol particles in SAM were simulated using the TwO Moment

Aerosol Sectional (TOMAS; Adams and Seinfeld, 2002) microphysical scheme embedded into SAM.
The algorithm simulated the size distribution across 13 logarithmically spaced size bins spanning 3 nm-
1 μm with 2 additional bins spanning 1-10 μm. The aerosol size distribution was tracked via two
independent moments for each bin of the size distribution (mass and number). TOMAS calculated
coagulation explicitly in each grid cell assuming a Brownian diffusion kernel (Seinfeld and Pandis,
2006). Our SAM-TOMAS simulations included only coagulation, and particles were assumed to be a
single species (no differentiating between BC and OA). The SAM-TOMAS model had previously been
tested against observations in Stevens et al. (2012) and Lonsdale et al. (2012) for power plant plumes.

We set background aerosol concentrations to zero as the biomass-burning aerosol

concentrations emitted into SAM-TOMAS were orders of magnitude larger than those present in a
remote background location, and as such the lack of background aerosol would have had an
insignificant effect on the rate of in-plume coagulational processing. In cases where the plume dilutes
to similar concentrations to the ambient background, subgrid-plume coagulation schemes are no longer
necessary, and grid-resolved coagulation will properly account for coagulation. The biomass-burning
aerosol was assumed to have a constant density of 1400 kg m$^{-3}$ as primarily a mix of organic
compounds, thus we do not consider how changes in BC/OA composition may affect density and
coagulation rates. The hygroscopicity of the aerosol particles was set to zero, allowing no water uptake.
This assumption is not true of real world biomass-burning aerosol and has been characterized in other
works finding hygroscopicities of fresh (κ=0.02-0.8; Petters et al., 2009) and aged smoke (κ=0.1-0.3;
Engelhart et al., 2012) with a strong dependence on fuel type. In terms of their effect on the size
distribution, a constant κ across all particle sizes has the simple effect of increasing the effective
diameter of the particles via water uptake by a scalar factor. This initial increase should only have a
relatively minor effect on the final dry $D_{pm}$ or σ of the plume after coagulational processing as the mean
coagulation rates are relatively insensitive to the size shifting of a particle population (Seinfeld and
Pandis, 2006; Stuart et al., 2013).

We ran 100 SAM-TOMAS simulations at 500 m x 500 m horizontal resolution (total cross-wind

(y-direction) horizontal extent = 100 km), and constant 40 m vertical resolution (total vertical extent =
4 km). This resolution accommodated the chosen plume parameters (see Sect. 2.1). The model was run
with a master timestep of 2 seconds (varied internally for accuracy in the coagulation calculation) for a
duration of 5 model hours (300 minutes). The output from each SAM-TOMAS simulation was
recorded at four different times (400 total time slices across 100 simulations) as the plume progressed
along the with-wind (x-direction) axis.

The seven inputs to the SAM-TOMAS model were constrained to capture a range of biomass-

burning characteristics in realistic scenarios and are summarized in Table 2. The ranges of values used
for $D_{pm0}$, $\sigma_0$, fire area and mass flux are the same as those listed in Table 1. The meteorological fields
were supplied by NCEP reanalysis meteorology from over North America (land only, lat: 30º - 70º N,
lon: 70º -135º W) during the July 2010 fire season. The SAM-TOMAS wall speed was set equal to the
mean boundary layer wind speed from NCEP. We filtered these inputs by requiring wind speed > 2 m s$^-$
$^1$ to eliminate stagnation situations over the source. The injection height (lower bound) and injection
depth of the aerosol were specified at between 50-1500 m and 500-2000 m respectively. No emission
injection parameterization (e.g. Freitas et al., 2007) was used as we were only trying to capture a range
of mixing depths for our aging calculation, and the absolute height was relatively unimportant. All the
SAM-TOMAS simulation inputs were chosen using semi-random Latin hypercube sampling across the
ranges listed above (Lee et al., 2012). The results of the full SAM-TOMAS simulation set are
summarized in Sect. 3.1.

We calculated the time-dependent mixing depth of the plume from vertical profiles averaged

horizontally across the entire simulation wall at each time slice. Figure 3 shows a sample of two
vertical profiles from different SAM-TOMAS simulations. The mixing depth was defined as the range
of altitudes where the aerosol mass was greater than half of the peak aerosol mass:
$$\text{mixing depth} = \Delta_{\text{alt 50\% peak aerosol mass}}$$

In cases where the plume mixed down to the ground, the lower altitude bound was defined as 0

m. Runs with mixing depths greater than 2500 m were excluded to ensure that the plume did not reach
the model top. In addition to mixing depth, $D_{pm}$ and $\sigma$ were calculated for each of the SAM-TOMAS
time slices from the first and third integrated moments of the size distribution as detailed by Whitby et
al. (1991).

We do not address new-particle formation in biomass-burning plumes in this work. In plumes

where new-particle formation in biomass-burning plumes occurs, our parameterizations will
underestimate the number of particles and overestimate the mean diameter of the plume particles.

## 2.3 Emulation of the SAM-TOMAS output

As running the full SAM-TOMAS model is too computationally expensive for implementation in global aerosol models, we built an offline emulator of the model for use as a parameterization in these global models. We created the emulator using the Gaussian Emulation Machine for Sensitivity Analysis (GEM-SA) developed by the Centre for Terrestrial Dynamics (http://www.ctcd.group.shef.ac.uk/gem.html). The GEM-SA software uses a Gaussian process to design a SAM-TOMAS simulator (the emulator) based on the behavior of the known SAM-TOMAS inputs and outputs (the training data). A complete description of GEM-SA statistics and assumptions can be found in Kennedy and O'Hagan (2001) and Kennedy et al. (2008). A description of its application as an estimator in atmospheric-aerosol modelling can be found in Lee et al. (2011). This software was previously used in sensitivity studies in atmospheric-aerosol (Lee et al., 2011, 2012) and vegetation models (Kennedy et al., 2008).

We used 400 data points from the set of 100 SAM-TOMAS simulations to train the emulator. GEM-SA assumes that the outputs are a continuous and differentiable function of the inputs to statistically emulate the model and estimate the SAM-TOMAS output ($D_{pm}$ and $\sigma$). We used a new set of completed SAM-TOMAS simulations (624 non-training data points) to test our GEM-SA parameterization for accuracy relative to SAM-TOMAS (see Sect. 3.2-3.3).

The GEM-SA parameterization requires seven input parameters: $D_{pm0}$, $\sigma_0$, mass flux, fire area, wind speed, mixing depth and time, and generates predicted aged $D_{pm}$ and $\sigma$ as outputs. These estimated $D_{pm}$ and $\sigma$ describe an aged lognormal aerosol mode incorporating the sub-grid scale coagulation taking place inside concentrated biomass-burning plumes and can be used in global/regional models. We have made the GEM-SA parameterization (emulator Fortran subroutine and input files) available as Supplementary Material.

# 3. Results

## 3.1 SAM-TOMAS simulation output

Figure 4 shows the $D_{pm}$ (panels a and c) and $\sigma$ (panels b and d) as a function of distance for each of the 100 SAM-TOMAS simulations used to train the emulator (Sect. 3.2). The influence of several factors (the distance from the source, emissions mass flux, and fire area) on the final aerosol size distributions

is apparent in the output of SAM-TOMAS simulations. Panels a and b are colored by the emissions
mass flux, whereas panels c and d are colored by dM/dxdz (kg m$^{-2}$, the amount of aerosol mass in an
infinitesimally thin slice of air perpendicular to the direction of the wind, i.e. mass flux · fire area /
wind speed/mixing depth). All simulations showed $D_{pm}$ increasing with distance as coagulation
progressed in each plume. The coloring in panel a shows that $D_{pm}$ generally increases more rapidly and
to higher values with higher emission fluxes. However, panel c shows that dM/dxdz appears to be a
better predictor for the increase of $D_{pm}$ with distance than the emissions flux, and the distance and
dM/dxdz capture much of the variability in $D_{pm}$.

Panels b and d show that σ tends to converge with distance as simulations with large initial σ

generally decrease with distance more rapidly than simulations with smaller initial σ. This convergence
happens slowly relative to the times simulated, so the initial σ have a strong influence even at 200 km.
The colors and panels b and d show that σ in high emissions-flux and dM/dxdz cases converge more
rapidly than low-emissions cases. However, as opposed to the 1.32 modal-width asymptote in the limit
of infinite coagulation found by Lee (1983), the SAM-TOMAS simulations converge to a limit of 1.2-
1.25. This is likely due to the size-distribution bin-spacing in the SAM-TOMAS model, where modal
widths <1.32 are smaller than a single TOMAS size bin width, which results in less accurate fits of σ
for smaller σ values.

Figure 5 is a scatterplot of σ vs $D_{pm}$ for each point seen in Fig. 4, excepting those at distances

less than 25 km (points close to the emissions source have been removed). The points are colored by
dM/dxdz. Thus, Fig. 5 shows the results of Fig. 4 panels c and d together but removes the distance
information. At these distances over 25 km, $D_{pm}$ is relatively well constrained by dM/dxdz alone,
showing that the mean growth by coagulation is strongly influenced by the mass of particles in the slice
of air. On the other hand, σ is unconstrained at low values of dM/dxdz but more constrained towards
1.2-1.4 at high values of dM/dxdz. At high dM/dxdz values, the convergence towards the steady-state σ
proceeds much more rapidly than at low dM/dxdz as also shown in Fig. 4d.

These SAM-TOMAS results show that dM/dxdz is a powerful determinant of aged biomass-

burning size. In these tests, we also explored the suitability of dM/dx (mass flux · fire area/ wind speed)
and dM/dV (initial mass concentration). Large mixing depths dilute particle concentrations and reduce
coagulation, so we expected that dM/dxdz may be a better predictor of biomass-burning size-
distribution aging than dM/dx. However, Fig. 4 and Fig. 5 did not look qualitatively different when
using dM/dx or dM/dV. A comparison of dM/dx vs dM/dxdz vs dM/dV in predicting final size-
distribution attributes is further discussed in Section 3.4. We quantitatively evaluate the fidelity of
dM/dx and dM/dxdz as proxies for biomass-burning size-distribution aging in Sect. 3.4. In the
following two subsections, we use the emulator to determine the contribution of the individual inputs to
the changes in simulated $D_{pm}$ and $\sigma$.

### 3.2 Model parameterization evaluation


We tested the GEM-SA-derived emulator parameterization against additional SAM-TOMAS model
runs that were not used in the fitting of the parameterization, and we show the results in Fig. 6. We use
624 additional SAM-TOMAS-simulated data points that were not used for GEM-SA training in this
evaluation. The emulator parameterization-predicted outputs corresponding to these data points for $D_{pm}$
and $\sigma$ are plotted against the SAM-TOMAS $D_{pm}$ and $\sigma$. Predicted $D_{pm}$ has an $R^2$ value of 0.83 with a
slope of 0.92. Larger absolute errors in $D_{pm}$ are found at the larger diameter sizes, but 86% are found
within 10% of the SAM-TOMAS $D_{pm}$ (76% of predicted $D_{pm}$ are within 5% of SAM-TOMAS $D_{pm}$).
The small mean normalized bias (MNB) of -0.06 indicates a slight negative bias in the
parameterization. This bias is generally seen towards the higher final $D_{pm}$ values in the simulations
(>250 nm), which are reached only by the most aged plumes with the heaviest aerosol loads. The $\sigma$ plot
(Fig. 6b) shows a similar correlation coefficient ($R^2$=0.91) and has a slope of 0.93. The MNB is 0.01
and 77% of the predicted $\sigma$ points are within 5% of the $\sigma$ calculated from SAM-TOMAS. The cluster of
points near $\sigma$ =1.2-1.3 is indicative of the modal width steady-state limit. This limit is not captured by
the $\sigma$ parameterization, which assumes a smooth function towards even lower $\sigma$ values.

### 3.3 Sensitivity of aged size distribution to input parameters


Figures 7 and 8 show the sensitivities of the parameterization outputs ($D_{pm}$ and $\sigma$, respectively) to the
input parameters ($D_{pm0}$, $\sigma_0$, mass flux, fire area, wind speed, time, and mixing depth) as determined by
the GEM-SA emulation of the SAM-TOMAS output. (Note that distance was used as the dependent
variable in Fig. 4, while we use time in the emulator. Time can be converted to distance by multiplying
by the wind speed). In every panel, each line shows the change in $D_{pm}$ (Fig. 7) or $\sigma$ (Fig. 8) as an input
parameter (e.g. $D_{pm0}$ in panel a) is varied systematically from its minimum to maximum tested value
with a randomly chosen set of the other six input parameters. Each panel contains 100 lines, which
means that 100 sets of the six other input parameters were randomly chosen to make these lines. We
normalize each line by the value of $D_{pm}$ or $\sigma$ at the midpoint of the x-axis (i.e. where the input
parameter is at the midpoint of its tested range). For time since emission (panel f) we normalize by the
values at t=0 min instead of at the midpoint of the range. These plots therefore show the percent change
in $D_{pm}$ or $\sigma$, $\Delta\%_{output}$, as each input is changed from its midpoint value (or t=0 min for time), in order to
emphasize the parameterization's output response to each isolated input variable.

The $D_{pm}$ sensitivity plots (Fig. 7) show a number of well-defined responses of $D_{pm}$ to the inputs.

$D_{pm}$ increases monotonically with increases in mass flux and fire area (Fig. 5b,d), and decreases nearly
monotonically with wind speed. These trends are due to the interrelationships of these inputs with
starting number concentration. These results are consistent with Fig. 4 and Fig. 5, where $D_{pm}$ increased
with increasing dM/dx in the SAM-TOMAS simulations. Additionally, the $D_{pm}$ also decreases
monotonically with mixing depth (albeit more weakly than mass flux, fire area, and wind speed), so
dM/dxdz may also be a good proxy for biomass-burning size-distribution aging (evaluated in Sect. 3.4).
Higher dM/dx and dM/dxdz values lead to higher initial number concentration in these plumes, which
drive higher rates of coagulation due the squared dependence of coagulation rate on number
concentrations.

$D_{pm}$ also increases nearly monotonically with time (the regions of slight decreases with time

show that the parameterization is not necessarily always physically representative due to the statistical
nature of the fit over the parameter space). The rapid rise in $D_{pm}$ for time <2 hrs is due to the high
number concentrations (N) and coagulation rates near the source. As dilution and coagulation progress,
N decreases and coagulation slows, resulting in a slowing of $D_{pm}$ increase. Mass flux has the largest
range of output $D_{pm}$ associated with the input ranges specified here ($\sim$ -50% to +100%).

The relationship between $D_{pm}$ and the initial size parameters ($D_{pm0}$ and $\sigma_0$) is more complicated.

Neither $D_{pm0}$ nor $\sigma_0$ show monotonic increases or decreases in $D_{pm}$ due to changes in either of these
isolated inputs. In general, there is an increasing trend in output $D_{pm}$ with increasing $D_{pm0}$, but for some
cases it decreases. These decreases in $D_{pm}$ are likely due to (1) decreasing particle number
concentrations with increasing $D_{pm0}$, which leads to reduced coagulation rates and (2) imperfections in
the statistical fit of the parameter space. The larger $\sigma_0$ indicate broader emission size distributions, with
more large particles and small particles. Since coagulation progresses fastest between large and small
particles (as opposed to particles of approximately the same size), this favors higher $D_{pm}$ at higher $\sigma$.
However, the initial particle number decreases with increasing $\sigma$, which lowers the coagulation rate and
leads to lower $D_{pm}$.

The emulator-derived $\sigma$ sensitivities are shown in Fig. 8. Since we expect $\sigma$ to converge towards

an asymptotic limit with coagulational processing (Fig. 4b,d), we see with those input parameters
associated with higher plume number density (mass flux, fire area, wind speed$^{-1}$, mixing depth$^{-1}$), which
gave monotonic increases for $D_{pm}$, show mixed results for $\sigma$ due to variability in the initial $\sigma_0$. The time
sensitivity plot (Fig. 8f) shows decreasing $\sigma$ with time similar to Fig. 4b,d.

Emission $\sigma_0$ shows the most pronounced and largest magnitude effect on output $\sigma$ ($\sim$ -30% to

+30%). Thus, the timescales for $\sigma$ evolving towards 1.2 is longer than the timescales tested here for
even the densest plumes. These sensitivity plots show that there is less variability in $\sigma$ than in $D_{pm}$ over
the tested input space.
*3.4 Simplified fits to the aged size distributions*
In addition to the GEM-SA emulator fits, we determined simplified fits for both $D_{pm}$ and $\sigma$ based on the
behavior in Fig. 4 and Fig. 5. These fits are easier to implement in regional and global aerosol models
than the full GEM-derived parameterization. These equations are meant to produce approximate
estimates of $D_{pm}$ and $\sigma$ throughout plume size-distribution aging. The equations require: the initial
value of the size-parameter of interest ($D_{pm0}$ or $\sigma_0$), a value proportional to the plume aerosol loading
(dM/dxdz: mass flux · fire area / wind speed / mixing depth or dM/dx: mass flux · fire area / wind
speed), and time since emission from the source fire (time). (Distance may also be used in these
equations rather than time, and distance/wind-speed should be used in place of time.) The functional
forms fitted for $D_{pm}$ and $\sigma$ are found below.

$$D_{pm} = D_{pm0} + A\,[dM/dx]^b\,(time)^c \tag{1}$$

$$D_{pm} = D_{pm0} + A\,[dM/dxdz]^b\,(time)^c \tag{2}$$

$$\sigma = \sigma_0 + A\,[dM/dx]^b\,(time)^c\,(1.2 - \sigma_0) \tag{3}$$

$$\sigma = \sigma_0 + A\,[dM/dxdz]^b\,(time)^c\,(1.2 - \sigma_0) \tag{4}$$


where A, b and c are determined by fitting each equation to the SAM-TOMAS data. For these
empirical equations, the units of dM/dx are kg m$^{-1}$, dM/dxdz are kg m$^{-2}$, $D_{pm}$ is nm and time since
emission is min. It should be noted that the equations for $D_{pm}$ and $\sigma$ are designed to be independent of
each other (i.e. $D_{pm}$ is not dependent on $\sigma_0$), which differs from the GEM-SA emulator. The aerosol
loading parameter dM/dx was chosen based on the stratification seen in Fig. 4c and Fig. 5. dM/dxdz
was tested as well, as it incorporates the variance associated with mixing depth into the fit. The fit to
dM/dx rather than dM/dxdz may be advantageous because we expect mixing depth of the plume to be
one of the more uncertain parameters in an atmospheric model, and the $D_{pm}$ sensitivities to mixing
depth tend to be smaller than those to mass flux, fire area and wind speed in the GEM-SA emulator
(Fig. 7). The $\sigma$ fits introduce a fourth factor, $(1.2\text{-}\sigma_0)$, which represents the difference between the
SAM-TOMAS infinite-coagulation limit (Fig. 4b and d) and the initial modal width.

The scalar A, b and c variables were fit to the ensemble of SAM-TOMAS data. Their values are

summarized in Table 3. The fits were tested against independent SAM-TOMAS data in Fig. 9 ($D_{pm}$) and
Fig. 10 ($\sigma$). The simplified $D_{pm}$ parameterizations, as expected, are not as good a fit of the SAM-
TOMAS data as the GEM-SA emulator (Fig. 6). The fit statistics for the simple parameterizations are
as follows: $D_{pm}$(dM/dx): slope = 0.82, $R^2$ = 0.67, MNB= 0.003, $D_{pm}$ (dM/dxdz): slope = 0.98, $R^2$ = 0.77,
MNB= 0.008. The fit using dM/dxdz generally performs better than that with dM/dx. The simple $\sigma$ fit
also did not perform as well as the GEM-SA emulator with fit statistics of: $\sigma$(dM/dx): slope = 0.64, $R^2$=
0.78, MNB= 0.02 and, $\sigma$(dM/dxdz): slope = 0.65, $R^2$ = 0.79, MNB= 0.01). Thus, dM/dxdz fits do yield
better results than dM/dx (in particular for $D_{pm}$); however, a user may choose to use the dM/dx fit if the
mixing depth is unknown. We note that these fits are only valid within the parameter ranges shown in
Table 1. dM/dV was also tested as a parameter within these simplified parameterization, but did not
yield better agreements for either $D_{pm}$ or $\sigma$ than dM/dxdz despite incorporating an additional plume
parameter (initial plume y-extent). This is because dM/dxdz is the product of dM/dV and the initial
plume width; since wider plumes are less susceptible to dilution than narrower plumes, dM/dxdz
captures this plume-width effect while dM/dV does not.

### *3.5 OA production/loss*


One of the limitations of the coagulation-only parameterizations derived in this paper is that they do not
include the effects of potential condensation/evaporation of organic aerosol on the aged biomass-
burning size distribution. Both condensational growth and evaporative loss of OA has been observed
previously in chamber studies and the field due to OA production or evaporation from
dilution/chemistry (Cubison et al., 2011; Hecobian et al., 2011; Hennigan et al., 2011; Grieshop et al.,
2009; Ortega et al., 2013; Jolleys et al., 2015; Vakkari et al., 2014). Konovalov et al. (2015) has
emphasized the importance of OA simulation in modeling long-range (>1000 km) plume evolution.
Thus, in order to predict biomass-burning aerosol mass, and thus the aerosol size distribution, we must
understand how OA evolves in biomass-burning plumes.
Here we present a simple correction to our coagulation-only parameterizations to account for
in-plume OA production/loss, assuming that this production/loss is known. This correction assumes all
SOA condenses onto existing particles (no new-particle formation). Each parameterization presented in
this paper may be corrected to include OA production/evaporation using the corrections below. We
assume that the OA production or loss does not affect the coagulation rates or $\sigma$, but acts to increase the
final $D_{pm}$. These assumptions are imperfect as irreversible condensation (evaporation) decreases
(increases) $\sigma$; however, $\sigma$ is preserved during condensation or evaporation of semi-volatile material
(Pierce et al., 2011). Regardless, for the relatively small amounts of OA condensation/evaporation
considered here, the change in $\sigma$ and coagulation rates should be minor. For a factor of 25% growth in
diameter from SOA, which may be expected from for a factor of 2 increase in OA mass with a small
change in sigma, we expect coagulation rates to stay within about 10% (Seinfeld and Pandis, 2006).
For larger changes in OA mass (more than a factor of ~2) due to production/loss, our simple correction
will have uncertainties due to these assumptions. Our correction to the final $D_{pm}$ has the following
form:

$$D_{pm\,w/OA\,prod/loss} = D_{pm\,w/o\,OA\,prod/loss} \cdot \left( \frac{OAMass_{w/OA\,prod/loss} + BCMass}{OAMass_{w/o\,OA\,prod/loss} + BCMass} \right)^{1/3} \quad\quad (5)$$


where $D_{pm\,w/o\,OA\,prod/loss}$ is the final $D_{pm}$ from the coagulation-only GEM-SA emulator parameterization,
the biomass-burning aerosol OA mass (with and without additional production or loss) is in kg (per
particle or volume of air) and the BC mass is in kg (per particle or volume of air). Thus, for a doubling
of OA due to SOA production (one of the larger enhancements found in Hennigan et al., 2011),
particles that contain negligible BC will grow in diameter by 26% above the coagulation-only
predictions. If the particles contained 50% BC, then the diameter growth would only be 14%.
While these changes are expected to be on the large end for growth by SOA production, they
are significantly smaller than the ~200% variability in aged $D_{pm}$ due to coagulation over the range of
initial fire conditions (Fig. 7). For example, variations in wind speed, mass flux, and fire area alone can
independently cause variability in the aged $D_{pm}$ by a factor of 2 due to changes in coagulation rates
while variability in condensational growth appears to cause much smaller uncertainties (~25%) in the
aged $D_{pm}$. This indicates that although SOA condensational growth is certainly important in shaping
particle composition and total particle mass, it is not among the most dominant factors determining the
aged $D_{pm}$ compared to those fire-condition parameters controlling coagulational growth. It should be
noted, however, that the $D_{pm}$ growth attributed to OA condensation is not accompanied by a change in
particle number (additional OA mass is distributed among existing particles), whereas a similar
increase in $D_{pm}$ growth by coagulation only would have an accompanying decrease in particle number.
Thus, the changes to the aerosol size distribution and climatic influence of a size change due to
coagulation and condensation are different.

### 3.6 Estimating aged size distributions observed at the Mt. Bachelor Observatory

The simplified fits presented in Section 3.4 (equations 1-4) were tested against size distributions
measurements of biomass-burning plumes observed at the Mt. Bachelor Observatory (MBO) in Central
Oregon (43.98°N, 121.69°W, 2,764 m a.s.l.). MBO is a mountaintop site that has been in operation
since 2004 (Jaffe et al., 2005). An intensive campaign was performed during the summer of 2015 to
measure aerosol physical and optical properties of wildfire emissions (Laing et al., in prep). During this
campaign aerosol size distributions from 14.1 to 637.8 nm were measured with a Scanning Mobility
Particle Sizer (SMPS). Additional details about MBO and the sampling campaign can be found in
Laing et al. (in prep).

We identified eleven biomass-burning plumes during August (Table 4). Criteria for plume

selection was aerosol scattering > 20 Mm[-1] and CO > 150 ppbv for at least an hour, a strong correlation
($R^2$ > 0.80) between aerosol scattering and CO, and consistent backward trajectories indicating
transport over known fire locations. We calculated back-trajectories to determine fire locations using
the National Oceanic and Atmospheric Administration Hybrid Single-Particle Lagrangian Integrated
Trajectory (HYSPLIT) model, version 4 (Draxler, 1999; Draxler and Hess, 1997, 1998; Stein et al.,
2015) with Global Data Assimilation System (GDAS, 1º x 1º) data. The Mt. Bachelor summit is located
at ~1500 m amgl (above model ground level), so the back-trajectory starting heights of 1300, 1500, and
1700 m amgl were chosen (Ambrose et al., 2011). Fire locations were identified using Moderate
Resolution Imaging Spectroradiometer (MODIS) satellite-derived active fire counts
(http://activefiremaps.fs.fed.us/; Justice et al., 2002).
For the plume aerosol loading parameterization inputs in equations 1-4, we used Fire INventory
from NCAR (FINN) daily-averaged fire area and fire-emissions estimates (Wiedinmyer et al. 2011).
Multiple FINN data points in the same vicinity were combined based on the location of large-wildfire
incidents tracked by the National Interagency Fire Center (NIFC) (http://activefiremaps.fs.fed.us/). We
calculated the mass flux for the aerosol-loading estimates (dM/dxdz and dM/dx) using these FINN
OC+BC emissions (kg/day) and FINN fire area data ($km^2$). Mixing depth was defined as the mixing
depth at the source location of the fire in the Global Data Assimilation System (GDAS, 1º x 1º) data.
Wind speed was also extracted from GDAS data and was calculated as the average wind speed from the
ground to the defined mixing height. If no data were available, the mixing height and wind speed were
set to 660 m and 8.5 m/s based on the median value of the rest of the plumes. We assumed the emission
diameter ($D_{pm0}$) to be 100 nm, and we calculated σ using initial modal widths ($σ_0$) of 1.6, 1.9 and 2.4, to
be discussed later. We estimated the transport time from plume back-trajectories, and these values
ranged from 4.5 to 35 hours.
The measured and calculated size distribution diameter and modal widths for each plume at
MBO are summarized in Table 4. We calculated $D_{pm}$ and σ as the geometric mean diameter and
geometric standard deviation of the plume averaged size distribution as measured by the SMPS,
respectively. The plume-averaged size distributions may be influenced by non-biomass-burning
particles included along the trajectory from the wildfire. Plumes 1, 2, and 4 have bimodal distributions.
The second mode (Aitken mode) of these distributions are an example of influence from a non-biomass
burning source. These three bimodal distributions have inflated σ values, which will be addressed later.
Due to the large number of fires in Northern California and Oregon during the summer of 2015, some
of the plumes observed at MBO were influenced by more than one fire (e.g. Figure 11). For these
plumes, we calculated aged $D_{pm}$ and σ values for each fire area (black squares in Figure 11) and a
weighted average based on aerosol loading (dM/dx or dM/dxdz) was taken. Column 3 in Table 4
indicates how many fire areas were averaged for each plume.
Figure 12 shows the predicted aged $D_{pm}$ plotted against the observed values for both the dM/dx
and dM/dxdz forms of the simple parameterization. An initial $D_{pm0}$ of 100 nm was assumed. Equation 2
(using aerosol mass loading dM/dxdz) estimates $D_{pm}$ somewhat more accurately (y = 0.93x + 17.1, $R^2$ =
0.551) than Eqn. 1, which uses aerosol mass loading dM/dx (y = 0.62x +53.1, $R^2$ = 0.532). Over half
of the variability in the observed $D_{pm}$ was captured by the simplified fits. Thus, the simple
parameterizations show skill at predicting the aged $D_{pm}$ values relative to choosing a constant value of
aged $D_{pm}$ as is typically done in regional and global models.
Figure 13 shows the predicted aged $\sigma$ plotted against the observed values for both
parameterization forms. Both parameterizations do not predict modal width as well as $D_{pm}$ (Figure 12).
The calculated modal width changed significantly when using different emission modal-width values
($\sigma_0$). Janhäll et al. (2010) found the $\sigma$ of fresh biomass burning emissions to range from ~1.6 to 1.9.
When using a $\sigma_0$ of 1.6, we underestimated all of the $\sigma$ values. Using a $\sigma_0$ of 1.9, we improved the
estimation of aged $\sigma$ ranging from 1.4-1.6 (Figure 13a). The three higher measured $\sigma$ values are from
the bimodal plumes mentioned previously, which have larger $\sigma$ values than would be due strictly to the
biomass-burning plume. We found that using a $\sigma_0$ of 2.4 provided the best fit for all of the measured
plumes (Figure 13b), 2.4 being the max $\sigma_0$ value from Table 2. The $\sigma$ simplified fits using $\sigma_0 = 2.4$ have
statistics of: $\sigma(dM/dx)$: $y = 0.50 + 1.00$, $R^2 = 0.513$, and $\sigma(dM/dxdz)$: $y = 0.57 + 0.77$, $R^2 = 0.468$.
Thus, both parameterizations do not predict modal width as well as $D_{pm;}$ however, these
parameterizations do show skill relative to assuming a constant value of $\sigma$.
The results from the regional fires demonstrate that the parameterizations in Eqs 1-4 can be
successfully used to estimate aged biomass-burning size distributions in regional biomass-burning
plumes with transport times up to 35 hours with significantly better skill than assuming fixed values for
size-distribution parameters. More investigations of individual aged biomass- burning plumes,
specifically with one clear source, should be completed to fully characterize this parameterization.

# 4. Conclusions


We used the SAM-TOMAS large-eddy simulation model and an emulation technique to explore the
evolution of biomass-burning aerosol size distributions due to coagulation and build coagulation-only
parameterizations of this size-distribution evolution. We have also provided a simple correction to the
parameterization for cases with net OA production or loss. We used the SAM-TOMAS model to
simulate plume dispersion and aerosol coagulation. The SAM-TOMAS results show that the aged $D_{pm}$
can be largely described by dM/dx and the distance from the source (or time since emission). These
results also show that the aged $\sigma$ moves from $\sigma_0$ towards a value of 1.2 at a rate that depends on dM/dx.
The GEM-SA program was used to derive a $D_{pm}$ and $\sigma$ emulator parameterization based on the
SAM-TOMAS results. The parameterization requires seven input parameters: emission $D_{pm0}$, emission
$\sigma_0$, mass flux, boundary layer wind speed, fire area, plume mixing depth, and time since emission. The
predicted $D_{pm}$ and $\sigma$ can then be used as effective unimodal biomass-burning size-distribution
parameters in regional and global aerosol models.
The $D_{pm}$ parameterization showed the strongest sensitivities to those input parameters associated
with the extent of aerosol loading within the plume (mass flux, fire area, wind speed). Across the fire
area and wind speed ranges tested here, final $D_{pm}$ varied by ± 50%. Mass flux had the largest associated
$D_{pm}$ sensitivity across the tested values (-50% to +100%). These sensitivities were larger than those
associated with mixing depth (~ -20% to 20%) or the initial size-distribution parameters ($D_{pm0}$: ~ -25%
to 25%, $\sigma_0$: ~ 15% to -15%). The $\sigma$ parameterization showed a uniform decrease in $\sigma$ with time and
strong sensitivities to the emission $\sigma_0$ (-30% to 30%). This strong sensitivity to $\sigma_0$ can be attributed to
the inertia in $\sigma$ evolution in simulations with large modal widths and relatively small mass loading,
where $\sigma$ will not converge quickly to the coagulational limit (1.2).
The GEM-SA-derived parameterization performed relatively well against the SAM-TOMAS
model with a correlation of $R^2$=0.83, slope of m=0.92 and a low mean normalized bias of MNB=-0.06
for $D_{pm}$. The $\sigma$ parameterization has fit statistics of $R^2$= 0.93, slope= 0.91 and MNB= 0.01. The $\sigma$
parameterization was unable to capture the coagulational limit of 1.2 seen in the SAM-TOMAS results
and instead extrapolated to lower values. This 1.2 limit differs from the 1.32 $\sigma$ limit proposed by Lee
(1983) due to the bin-spacing in SAM-TOMAS being coarser than lognormal modes with these small
modal widths.
We also provided simplified polynomial fits for $D_{pm}$ and $\sigma$ (Eqns 1-4, Table 3) for calculating
aged $D_{pm}$ and $\sigma$ as independent functions of: the fresh emission parameter ($D_{pm0}$ or $\sigma_0$), the mass loading
of the aerosol (dM/dx or dM/dxdz) and the time since emission from the source fire. The $\sigma$ fits also
require a convergence term to account for the coagulational limit (1.2 in the SAM-TOMAS model).
Tested against independent SAM-TOMAS data, the $D_{pm}$ simplified fits performed as: $D_{pm}$(dM/dx):
slope = 0.82, $R^2$ = 0.67, MNB= 0.003 and $D_{pm}$ (dM/dxdz): slope = 0.98, $R^2$ = 0.77, MNB= 0.008. The $\sigma$
simplified fits have statistics of $\sigma$(dM/dx): slope = 0.64, $R^2$= 0.78, MNB= 0.02 and $\sigma$(dM/dxdz): slope
= 0.65, $R^2$ = 0.79, MNB= 0.01. The equations requiring (dM/dxdz) performed better than their (dM/dx)
counterparts as they also account for the aerosol layer depth.
We provided a correction for OA production/loss, and showed that significant production of
SOA within the plume (~ 100% OA mass enhancement) would cause a relatively small shift in the size-
distribution $D_{pm}$ (14-26% increase) compared to other factors that control the coagulation rate (e.g.
dM/dx). We note, however, that OA production increases $D_{pm}$ without loss of particle number while
coagulation increases $D_{pm}$ with a decrease in number, thus the climatic impact of condensation and
coagulation are different. The simplified OA-production/loss correction assumes no change in $\sigma$ with
condensational growth. Further testing should be done with explicit OA production and loss to better
quantify the effects of condensation of the size-distribution evolution.
We tested the simplified fits for $D_{pm}$ and $\sigma$ (Eqns 1-4, Table 3) against 11 aged biomass-burning
plumes observed at the Mt. Bachelor Observatory in August of 2015. $D_{pm}$ was reasonably calculated
using both measures of aerosol loading, dM/dx and dM/dxdz ($R^2$ values above 0.7 without an outlier).
The fit of calculated $\sigma$ and measured $\sigma$ depended heavily on the assumed initial modal width, with an
assumed $\sigma_0$ of 2.4 working best in our case ($R^2$ values around 0.75 without an outlier). Despite the
changes in calculated $D_{pm}$ and $\sigma$ due to the estimated emission size distribution, the parameterizations
captured the differences from plume to plume in regional biomass-burning plumes, which is based on
estimated aerosol loading and transport times.
Our analysis does not include any cloud processing of the plume particles, i.e. the production of
aqueous SOA within activated plume particles is not accounted for in our simple OA mass correction.
The production of SOA within droplets could result in additional SOA mass being only added to the
larger, activated particles during activation/evaporation cycling. This extra SOA mass would favor
increases in the diameters of the larger particles of the size-distribution only, which could create a
bimodal size distribution and increase the overall coagulational rates in the plume (more, larger
particles coagulate more rapidly with the small-diameter particles).
Future work includes (1) more testing of the parameterizations against real world observations
of size distribution aging, and (2) incorporating the parameterizations into regional and global aerosol
models for further evaluation against regional/global measurements.

# 5. Author Contribution

K.M. Sakamoto, R.G. Stevens, and J.R. Pierce designed the study. K. M. Sakamoto performed the
SAM-TOMAS simulations, and created and evaluated the parameterizations. J.R. Laing tested the
parameterizations size distributions of aged biomass burning plumes observed at the Mt. Bachelor
Observatory, and D.A. Jaffe oversaw the Mt. Bachelor measurements. K. M. Sakamoto prepared the
manuscript with assistance from all co-authors.

## 6. Acknowledgements

NCEP Reanalysis data provided by the NOAA/OAR/ESRL PSD, Boulder, Colorado, USA, from their
Web site at http://www.esrl.noaa.gov/psd/. K.M. Sakamoto was funded by a Natural Sciences and
Engineering Research Council of Canada (NSERC) PGS-M Fellowship. The authors gratefully
acknowledge the NOAA Air Resources Laboratory (ARL) for the provision of the HYSPLIT transport
model used in this publication

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

**Table 1.** Parameter ranges for each of the seven input parameters investigated in this study.

| Parameter | Description | Units | Min. Value | Max. Value |
|---|---|---|---|---|
| $D_{pm0}$ | Emission median dry diameter | nm | 20 | 100 |
| $\sigma_0$ | Emission modal width | - | 1.2 | 2.4 |
| Mass Flux | Emission mass flux from fire | $kg\ m^{-2}\ s^{-1}$ | $2 \times 10^{-8}$ | $5 \times 10^{-6}$ |
| Fire area | Square fire emissions area | $km^2$ | 1 | 49 |
| Wind speed | Mean boundary-layer wind speed | $m\ s^{-1}$ | 2 | 20 |
| Mixing depth | Mixing depth of aerosol layer | m | 150 | 2500 |
| Time | Time since emission | min | 0 | 300 |

**Table 2.** Parameter ranges for inputs to the SAM-TOMAS model.

| Parameter | Description | Units | Min. value | Max. value |
|---|---|---|---|---|
| Date | Req. for Met. field selection | 8-hour | July 1, 2010 | July 31, 2010 |
| Latitude | | deg N | 30 | 70 |
| Longitude | | deg W | 70 | 135 |
| $D_{pm0}$ | Emission median dry diameter | nm | 20 | 100 |
| $\sigma_0$ | Emission modal width | - | 1.2 | 2.4 |
| Mass Flux | Emission mass flux from fire | kg m$^{-2}$ s$^{-1}$ | $2 \times 10^{-8}$ | $5 \times 10^{-6}$ |
| Fire area | Square fire emissions area | km$^2$ | 1 | 49 |
| Injection height | Lower plume injection bound | m | 50 | 150 |
| Injection depth | Depth of plume at emission | m | 500 | 2000 |

**Table 3.** Best-fit parameters for the simplified $D_{pm}$ and $\sigma$ SAM-TOMAS parameterizations (Eqns. 1 to 4)

| Fit | Eqn. # | Parameter | | |
|---|---|---|---|---|
| | | A | b | c |
| $D_{pm}(dM/dx)$ | (1) | 4.268 | 0.3854 | 0.4915 |
| $D_{pm}(dM/dxdz)$ | (2) | 84.58 | 0.4191 | 0.4870 |
| $\sigma(dM/dx)$ | (3) | 0.05940 | 0.1915 | 0.3569 |
| $\sigma(dM/dxdz)$ | (4) | 0.2390 | 0.1889 | 0.3540 |

Table 4: Measured and calculated $D_{pm}$ and σ of biomass-burning plumes observed at MBO during August 2015. For the calculated $D_{pm}$ and σ of, the initial size parameters used were $D_{pm0}$ = 100 nm and $σ_0$ = 1.9.

| Plume | Plume date and time (UTC) | # fire areas | Measured (SMPS) | | Calculated | | | |
| | | | | | using dM/dx | | using dM/dxdz | |
| | | | $D_{pm}$ (nm) | σ | $D_{pm}$ (nm) | σ | $D_{pm}$ (nm) | σ |
|---|---|---|---|---|---|---|---|---|
| 1 | 8/9/2015  3:00-4:00 | 3 | 136.1 | 1.95 | 140.7 | 1.64 | 151.1 | 1.59 |
| 2 | 8/9/2015  5:00-7:00 | 3 | 144.0 | 1.77 | 140.8 | 1.64 | 152.0 | 1.58 |
| 3 | 8/10/2015  3:00-5:00 | 3 | 190.1 | 1.50 | 140.9 | 1.63 | 149.7 | 1.58 |
| 4 | 8/23/2015  3:55-7:00 | 1 | 162.5 | 1.89 | 145.5 | 1.63 | 162.4 | 1.57 |
| 5 | 8/24/2015  4:00-7:25 | 1 | 201.1 | 1.59 | 167.5 | 1.55 | 184.7 | 1.49 |
| 6 | 8/24/2015  7:30-11:20 | 1 | 217.5 | 1.52 | 190.1 | 1.50 | 230.1 | 1.40 |
| 7 | 8/24/2015  13:00-18:00 | 1 | 212.5 | 1.49 | 193.9 | 1.48 | 237.8 | 1.37 |
| 8 | 8/25/2015  3:50-6:50 | 1 | 192.2 | 1.54 | 161.4 | 1.57 | 172.6 | 1.52 |
| 9 | 8/27/2015  9:00-13:00 | 3 | 192.9 | 1.50 | 194.2 | 1.49 | 220.6 | 1.43 |
| 10 | 8/28/2015  8:00-11:15 | 3 | 183.4 | 1.54 | 182.1 | 1.50 | 203.2 | 1.43 |
| 11 | 8/28/2015  17:40-19:40 | 3 | 176.7 | 1.60 | 181.4 | 1.50 | 202.0 | 1.43 |

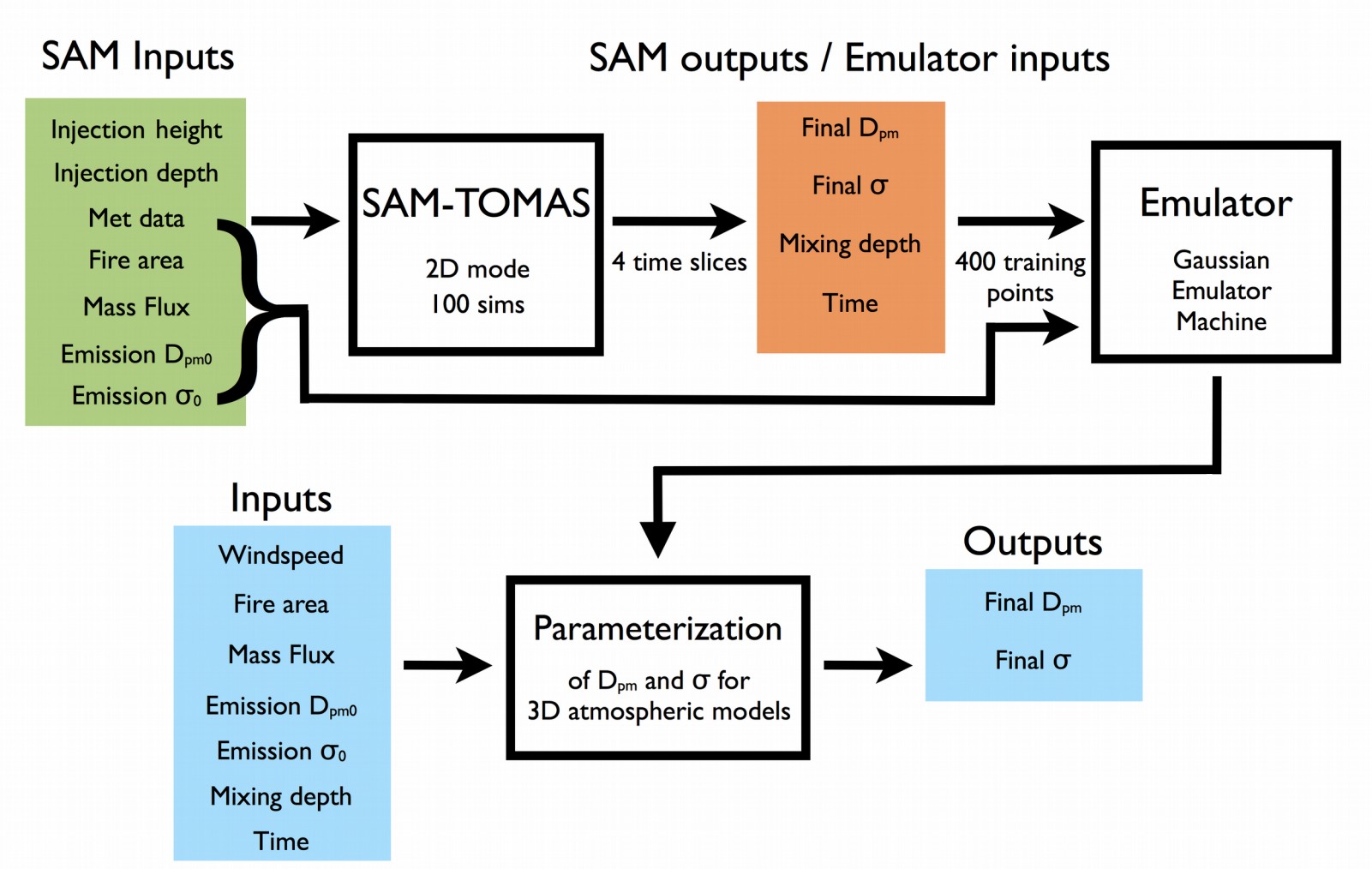

**Figure 1.** Schematic of the methods in this paper.

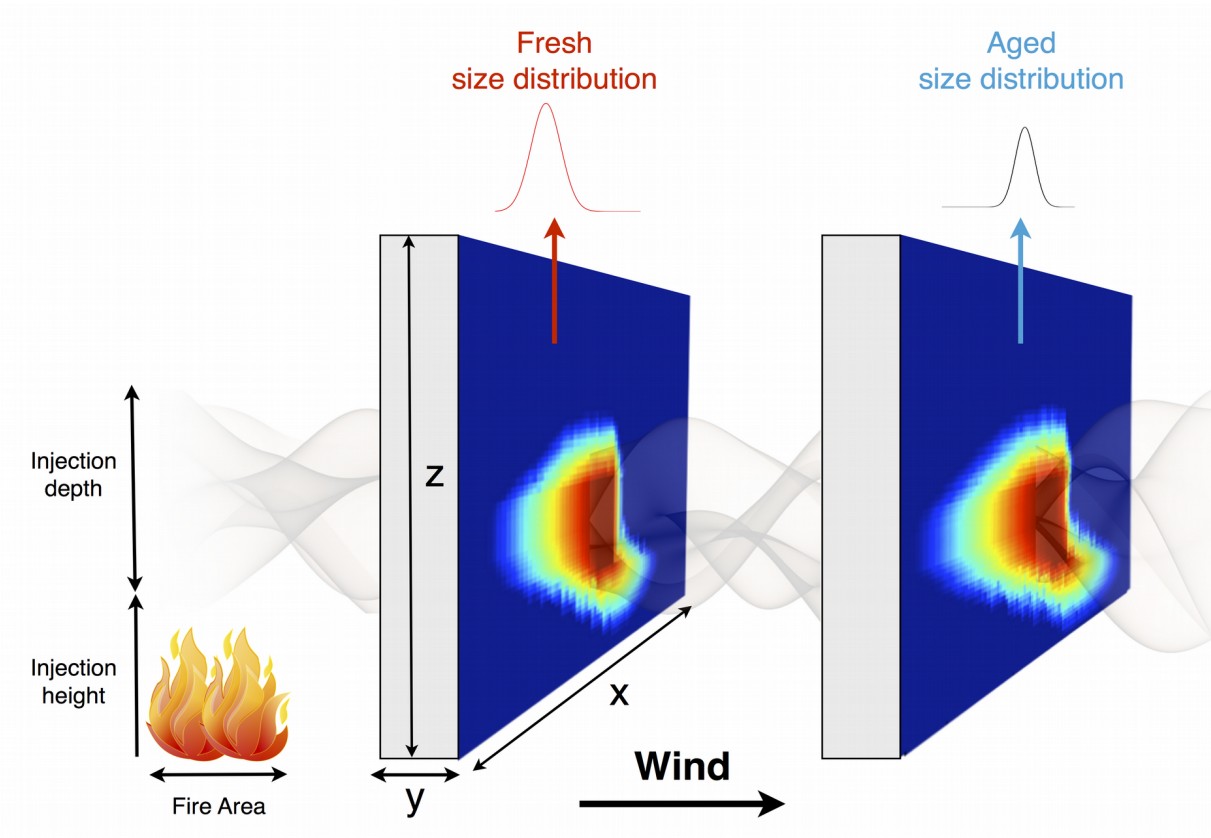

**Figure 2.** Schematic of a 2D SAM-TOMAS plume simulation.

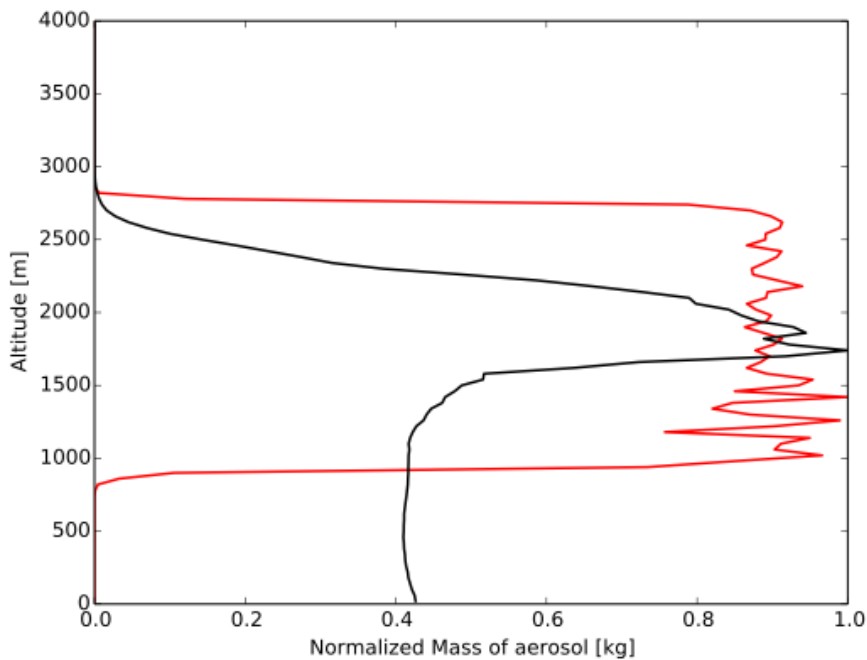

Figure 3. Final vertical profiles for two representative SAM-TOMAS simulations after four hours, normalized to individual aerosol load and averaged horizontally across the domain. The black profile shows a simulation where the aerosol mixed through the boundary layer to the ground with some aersol still trapped in a stable emission layer, while the red profile shows a simulation where the aerosol plume is still stable at the emission injection layer.

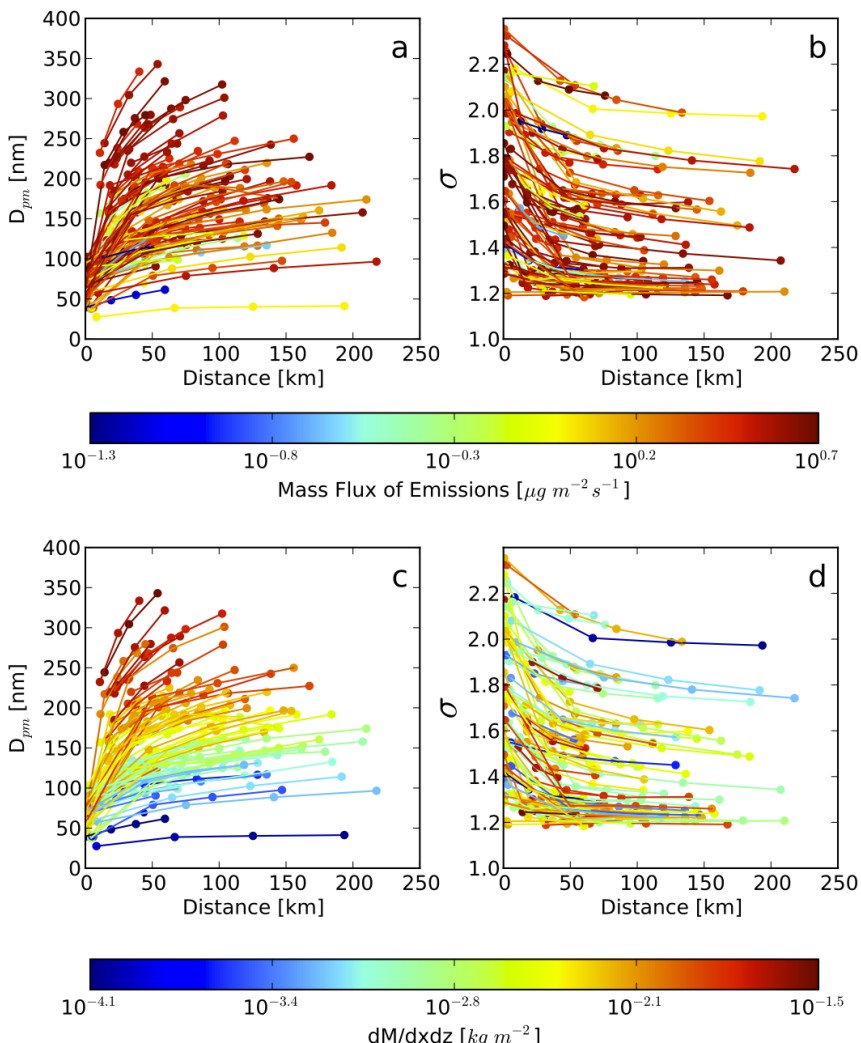

**Figure 4.** Wire plots showing size-distribution changes across individual SAM-TOMAS simulations colored by emission mass flux (panels a and b) and dM/dxdz (panels c and d) for $D_{pm}$ (panels a and c) and $\sigma$ (panels b and d).

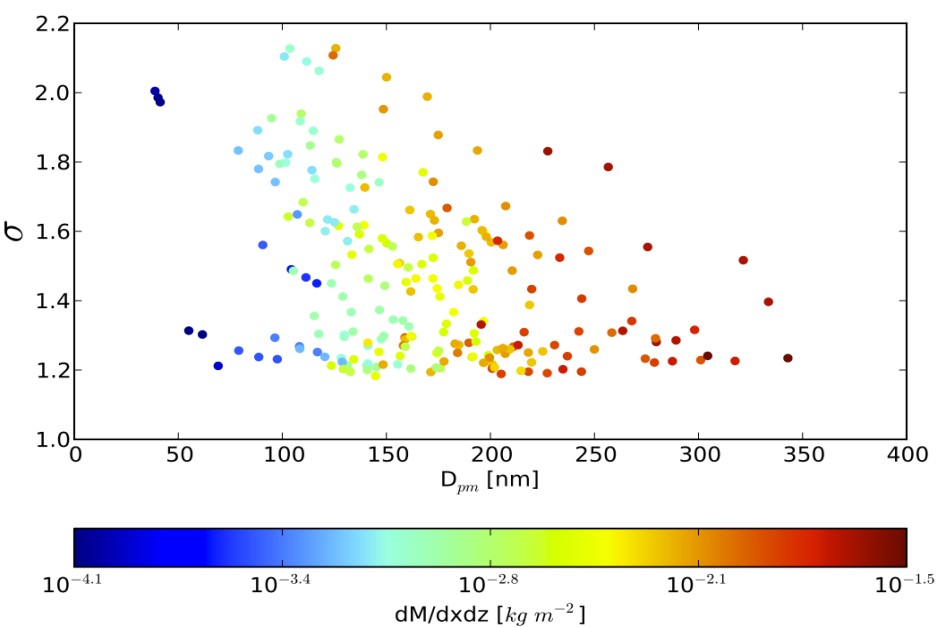

**Figure 5.** Scatter plot showing the relationships between final modal width (σ), final D$_{pm}$, and dM/dxdz for each of the SAM-TOMAS simulation slices at distances greater than 25 km from the fire.

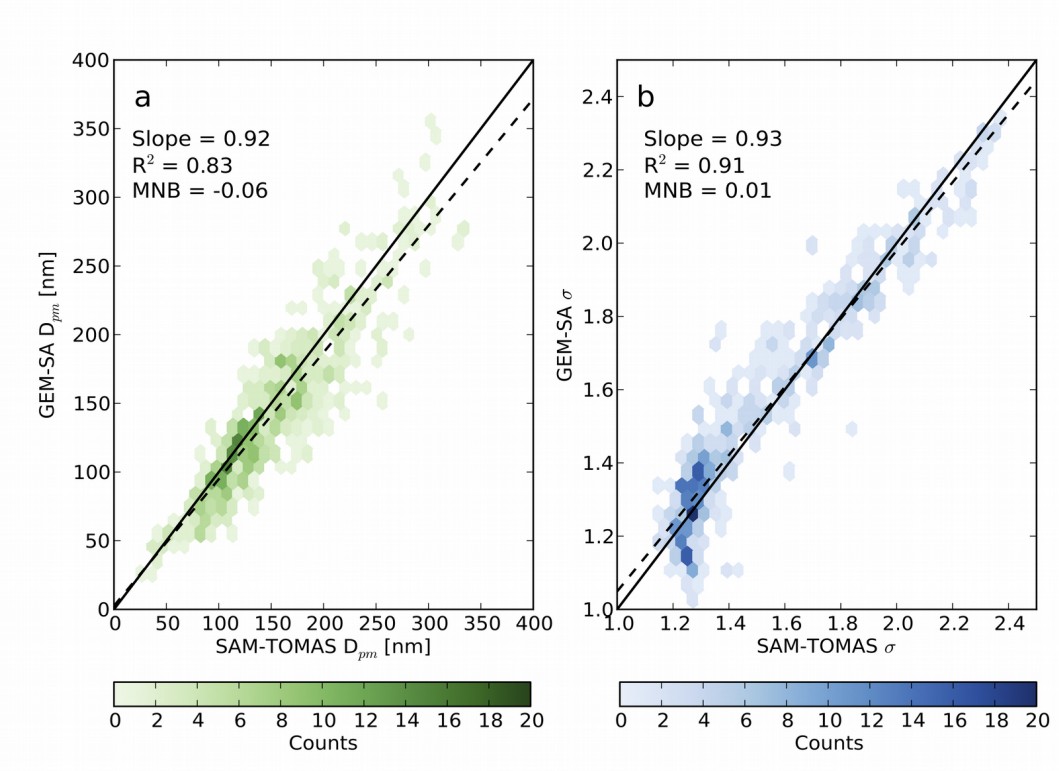

**Figure 6.** One-to-one plots showing GEM-SA emulator vs. SAM-TOMAS for 624 non-training simulation slices for a) final $D_{pm}$, and b) final modal width, σ. The black line is the one-to-one line. The dashed black line is the line of best fit.

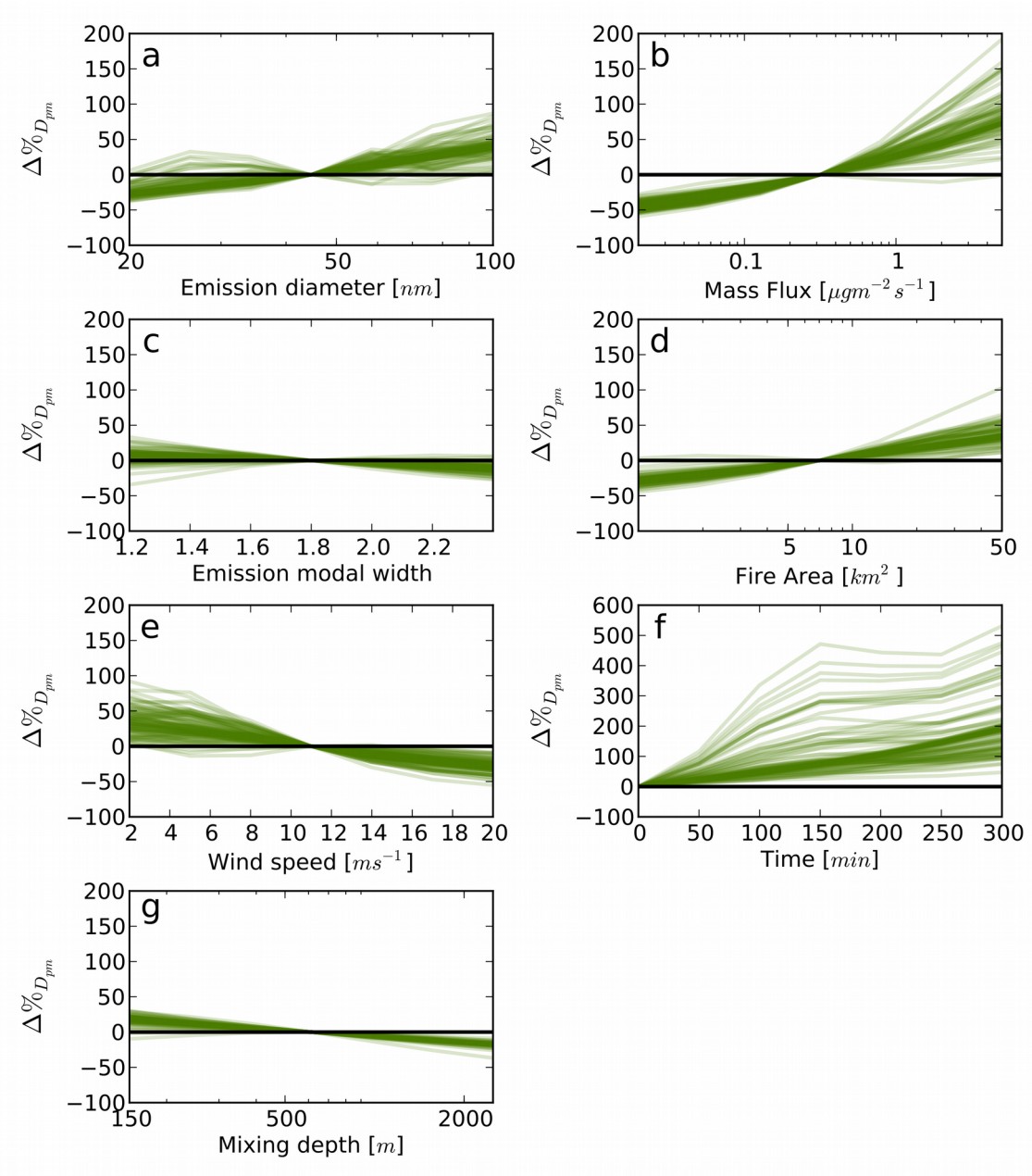

**Figure 7.** Sensitivity plots for the seven input parameters to the GEM-SA $D_{pm}$ parameterization. For each panel, a single input parameter is varied systematically from its minimum to maximum value for 100 randomly chosen sets of the other six parameters (100 lines in each panel). The sensitivities are shown as percent change in final $D_{pm}$, individually normalized to the value at the center of the x-axis (to zero in Time).

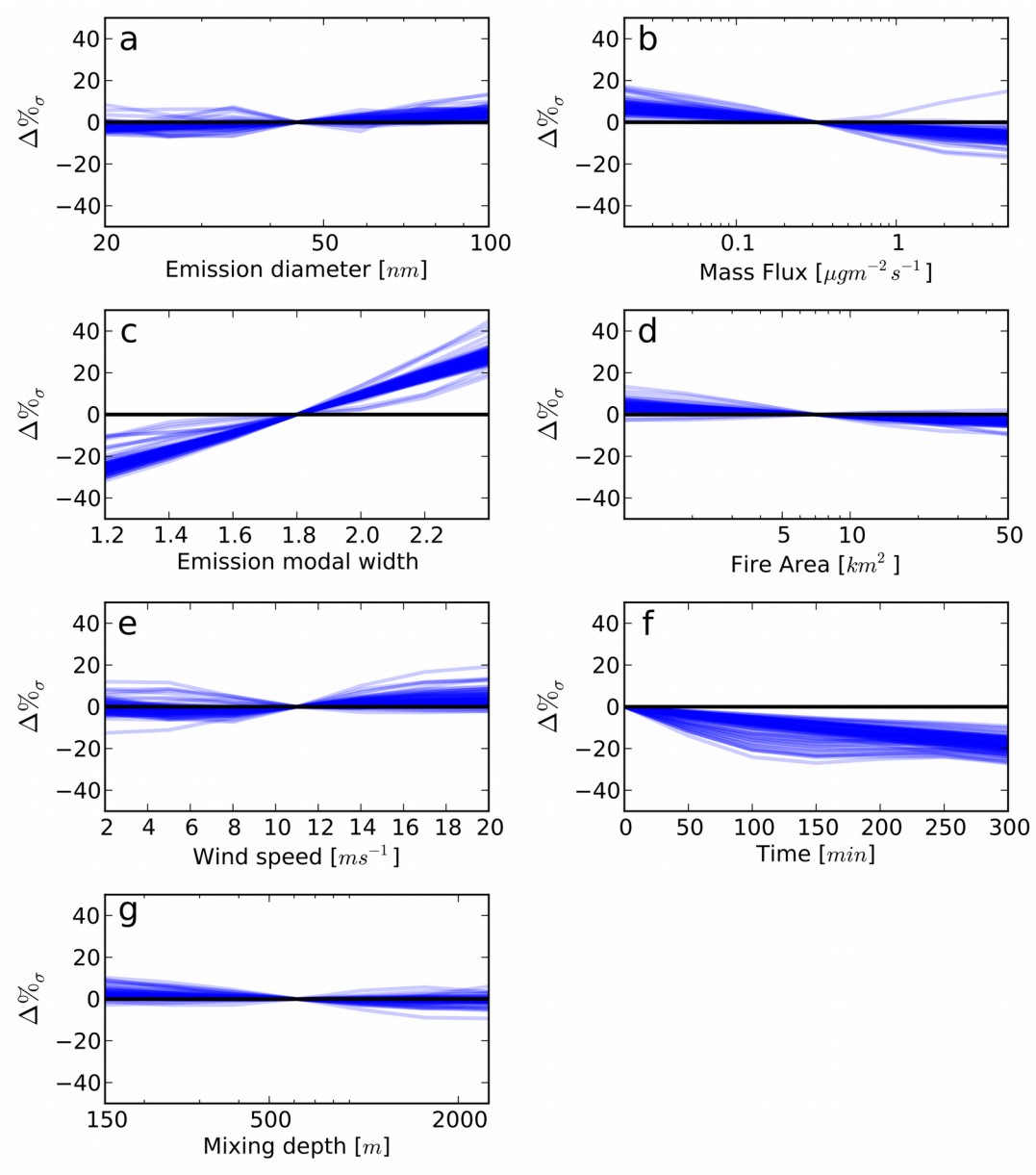

**Figure 8.** Sensitivity plots for the seven input parameters to the GEM-SA σ emulator parameterization. For each panel, a single input parameter is varied systematically from its minimum to maximum value for 100 randomly chosen sets of the other six parameters (100 lines in each panel). The sensitivities are shown as percent change in final σ, individually normalized to the center value of the x-axis (to zero in Time).

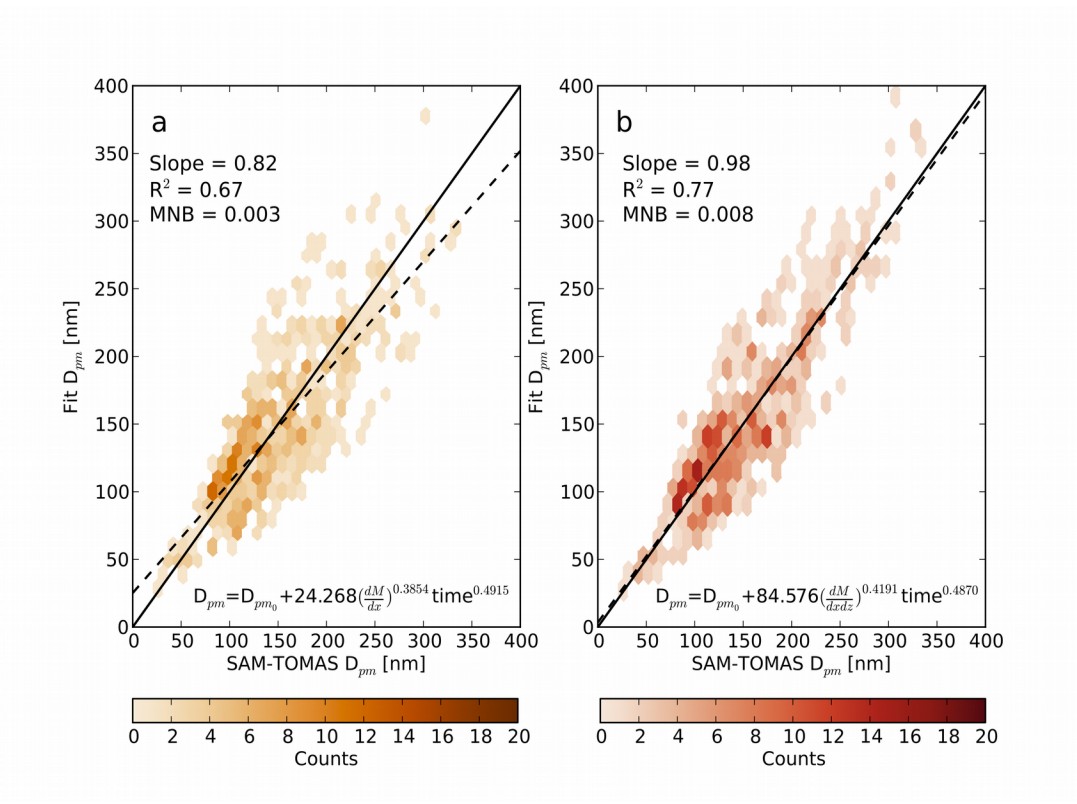

**Figure 9.** One-to-one plot showing simplified $D_{pm}$ fits vs SAM-TOMAS for a) dM/dx, and b) dM/dxdz. The black line is the one-to-one line. The dashed black line is the line of best fit. N = 624.

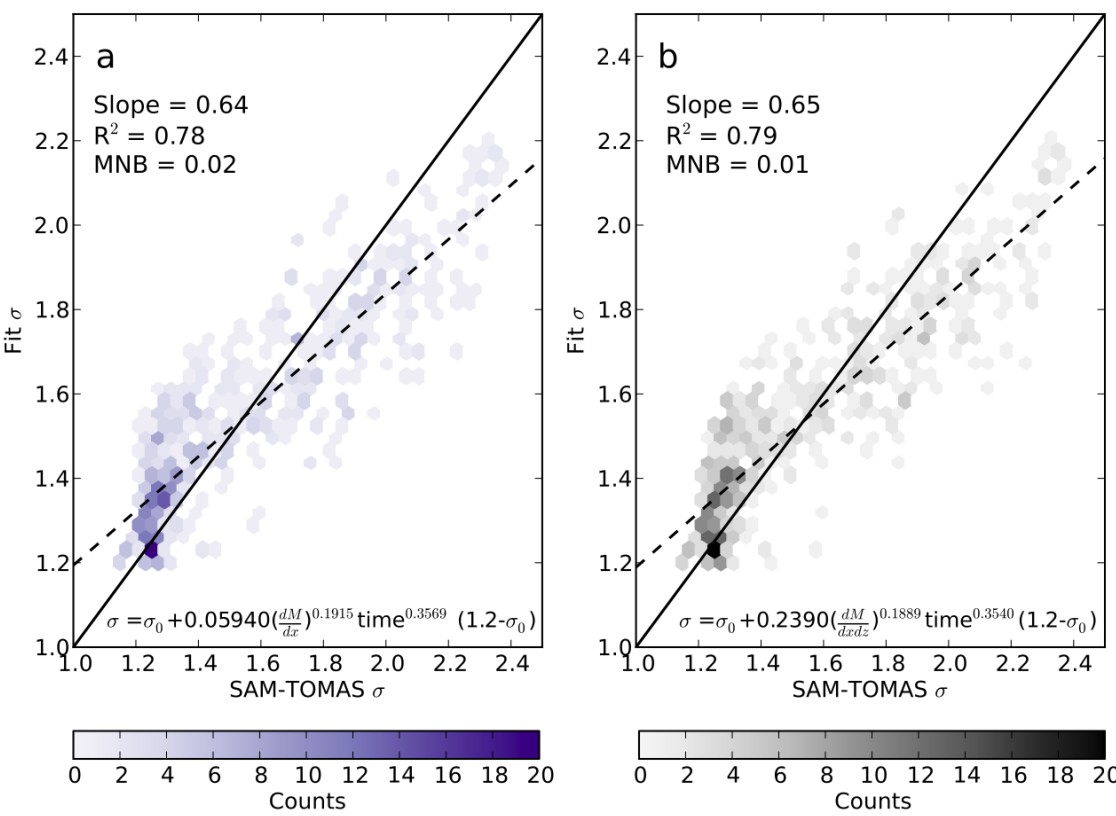

**Figure 10.** One-to-one plot showing simplified σ fits vs SAM-TOMAS for a) dM/dx, and b) dM/dxdz. The solid black line is the one-to-one line. The dashed black line is the line of best linear fit. N = 624.

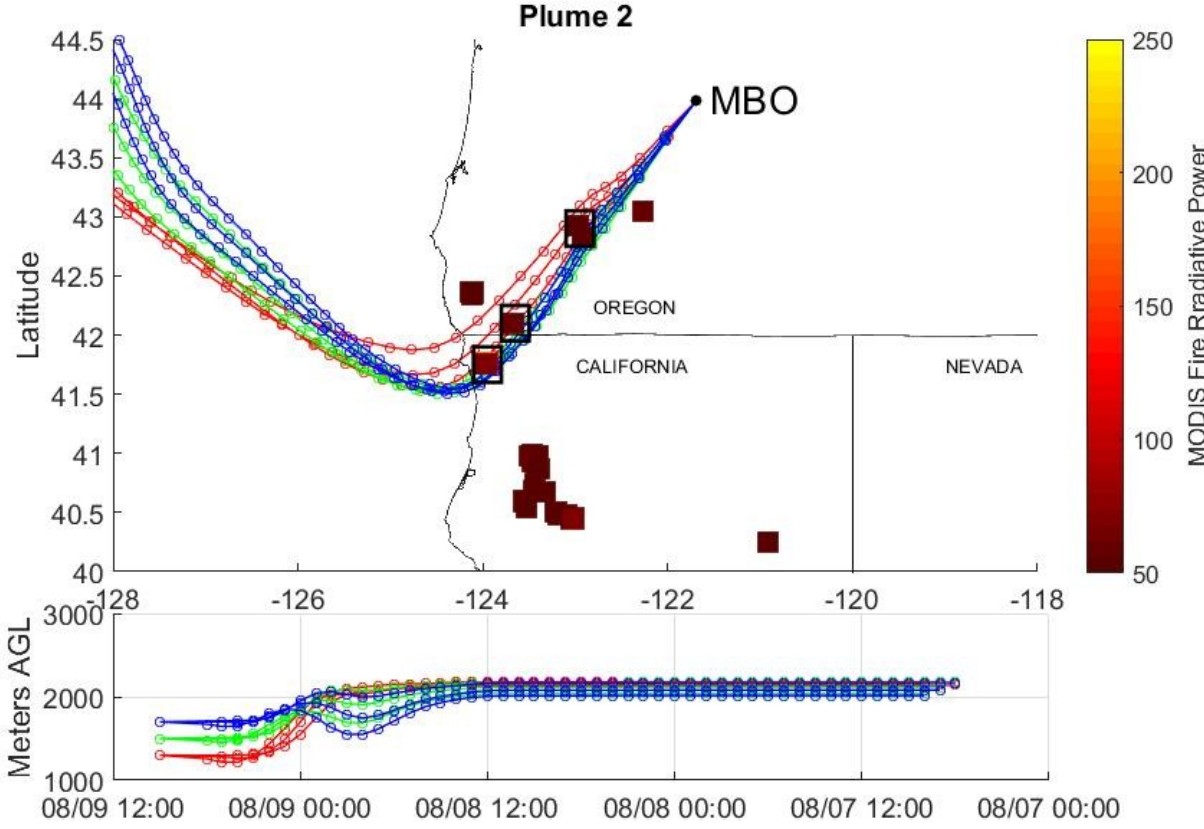

Figure 11. Back-trajectories from plume 2 observed at MBO. The colored squares represent fires during the time of the back-trajectory and are colored by Fire Radiative Power (FRP). The black squares indicate the fire areas used in the parameterization to estimate $D_{pm}$ and $\sigma$.

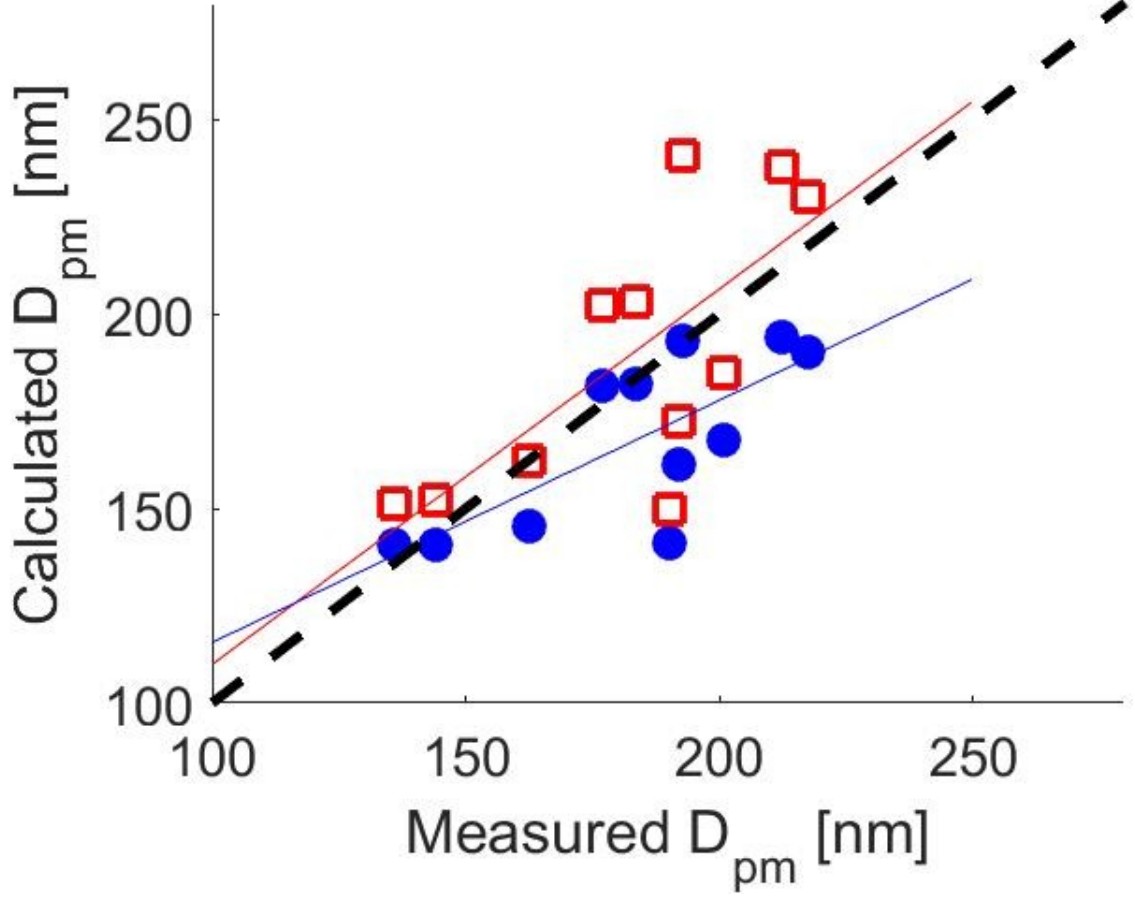

Figure 12. Scatter plot showing calculated and measured $D_{pm}$ for biomass-burning plumes observed at MBO during August of 2015. The blue circles represent $D_{pm}$ calculated using Eqn. 1 (dM/dx), and the red circles represent $D_{pm}$ calculated using Eqn. 2 (dM/dxdz).

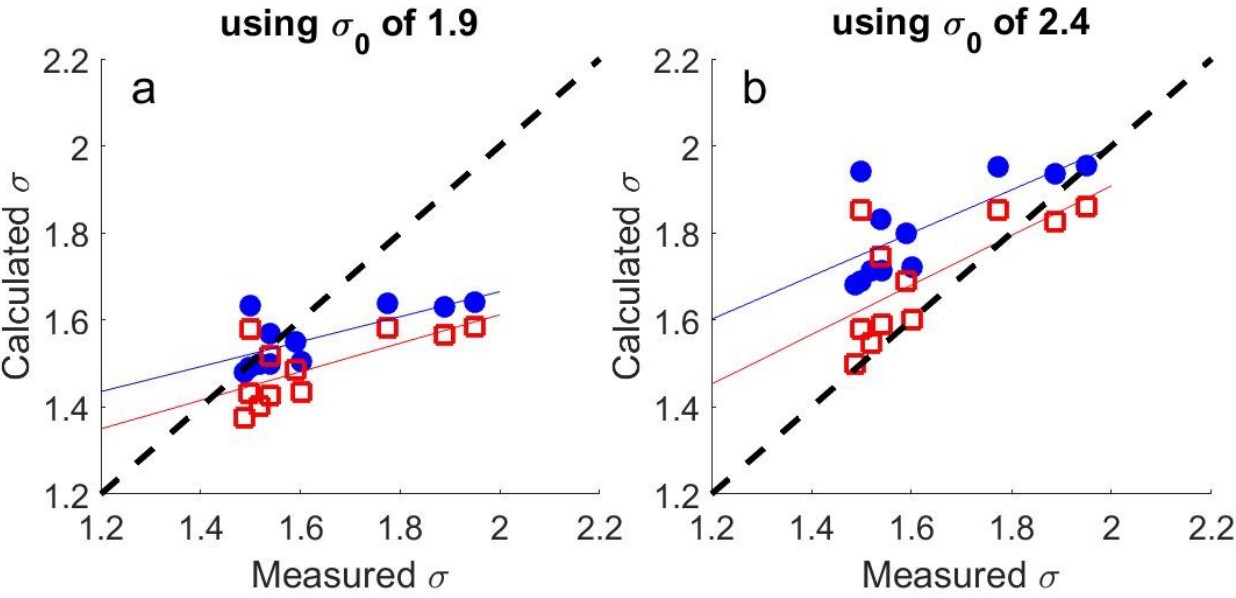

Figure 13. Scatter plots showing calculated and measured modal width (σ) for biomass-burning plumes observed at MBO during August of 2015. The blue circles represent σ calculated using Eqn. 3 (dM/dx), and the red circles represent σ calculated using Eqn 4. (dM/dxdz). Different emission modal width values ($\sigma_0$) were used to calculate σ, (a) used a $\sigma_0$ of 1.9 and (b) used a $\sigma_0$ of 2.4.