# Peer review of "The evolution of biomass-burning aerosol size distributions due to"

_Atmospheric Chemistry and Physics, 2016_

## Referee Comment (RC1) · Anonymous Referee #2 · 8 Mar 2016

This paper investigates the influence of coagulation on the particle number size distribution, notably on the mean diameter (Dm) and geometric standard deviation (Sigma) of a single particle mode, in biomass burning plumes. The work is based on a large number of sophisticated model simulations. The authors investigate how Dm and Sigma evolve with time in biomass burning plumes, and how their evolution is related to several parameters associated with primary particle emissions, fire conditions and atmospheric conditions. The authors compare briefly the influence of coagulation to that caused by organic aerosol formation/loss in a plume. The authors finally parameterize their results to a form that is applicable in large-scale modeling frameworks.

[Figure]

The is scientifically sound and orginal. The text is well organized and easy to read (with a couple of minor exceptions mentined below). The authors are able to exlain very well the numerous results obtained from model simulations. I have only a few minor suggestions for revisions.

Scientific comments:

I have a hard time of understanding Figure 3, even after reading the text on lines 210-213. I recommend that the authors work a bit more to make their message here clearer to readers.

Lines 254-255. The authors state that the initial mode mean diameter have little effect on Dm. I do not get this point when looking at Figures 4a and c: if Dm is initially large, it seems to typicaly lead to higher values of Dm at later plume times compared with cases when Dm is intially small. Could the authors specify what they mean here?

Line 390: Is this correct? Condensation of a non-volatile vapor into a single mode tend to narrow this mode, not widen it, as stated here.

The authors analyze shortly the influence OA production/loss on their results (section 3.5), and discuss also the potential effects of cloud processing (lines 463-469). This is clearly sufficient for these two processes in this paper. However, the authors do not mention at all new particle formation (NPF) that has been estimated to be a frequent process in biomass burning plumes. NPF might have notable effects on aerosol size distribution, and thereby on both Dm and Sigma, in evolving biomass burning plume. The authors should spend at least a few lines on discussing the relevance of this process in biomass burning plumes and on the potential effects of NPF on their results.

Technical issues:

Line 265: Figures 5 shows Sigma versus Dm rather than Dm versus Sigma.

Line 382: "..OA has been..."

---

## Referee Comment (RC2) · Anonymous Referee #1 · 23 Mar 2016

This paper studies the effect of coagulation on particle diameter (Dm) and geometric standard deviation (sigma) in biomass burning plumes using a large-eddy simulation model with an online aerosol microphysical module. The topic is timely and the text is well written; however, there are some issues that need to be clarified before this manuscript can be accepted. In my opinion this paper presents a valuable base case for assessing the role of coagulation and condensation in future biomass burning studies.

Major comments

I have two major concerns regarding this study: the lack of organic aerosol (OA) chemistry and that while interpreting the results little attention is placed on plume dilution and its effect on coagulation.

Running the simulations as "coagulation-only" limits the usability of the results. For instance, new particle formation has been observed in biomass burning plumes (Hennigan et al., 2012; Vakkari et al., 2014), as well as up to a factor of 4 mass increase during the first few hours (Vakkari et al., 2014). Also a recent study by Konovalov et al. (2015) suggests that accounting for OA volatility can improve model performance significantly, although over a much longer time scale than what is considered here. Therefore, without OA evaporation and condensation, can the parameterisations in this paper be a good starting point for global and regional scale models (c.f. line 37)?

On lines 453-455 it is concluded that SOA formation within the plume has a minor effect compared to coagulation. However, this is based on the assumption that SOA formation does not alter coagulation rate or sigma (lines 388-389). Could you elaborate on the conditions when these assumptions hold? For instance Pierce and Adams (2009) showed that secondary aerosol formation rate is one of the key parameters affecting how large fraction of small particles can grow up to CCN-sizes in new particle formation.

My second major concern is related to the effect of plume dilution on the coagulation rate. Coagulation depends strongly on aerosol particle number concentration (as stated on lines 145-146). However, the observed changes in Dm and sigma are not discussed in terms of concentration, but only with respect to the input parameters and a rather arbitrary dM/dx (aerosol mass in an infinitesimally thin slice of air perpendicular to wind direction). The effect of dilution on coagulation is mentioned only briefly (e.g. lines 324-326), though Figure 4 shows that in most simulations the Dm and sigma change rapidly near emission, but very slowly later on. Is this decrease in the rate of change in Dm and sigma due to plume dilution and subsequent slowing of coagulation rate?

How does Figure 4 look like if you colour it with concentration instead of dM/dx, or plot Dm and sigma against concentration? The dM/dx takes into account only dilution along the wind direction, not dilution due to vertical or cross-wind mixing.

Can you identify a range (time and space), where coagulation can cause significant changes in the size distribution and after which the plumes become so diluted that co-agulation slows down? How would this turning point depend on the initial concentration (emissions) and the meteorological conditions (turbulent mixing) during transport?

The background aerosol is assumed to be negligible compared to the plume and is set to zero (lines 176-179). However, in ambient air measurements this assumption cannot be made – see e.g. Yokelson et al. (2009). Have you verified that your plumes are so concentrated even after 200km transport that this assumption still holds? When will coagulation rate with the background aerosol become similar to coagulation within the biomass burning mode?

How is turbulent mixing handled in the simulations? Table 1 (page 23) lists "Mixing depth of aerosol layer" as an input parameter, yet on line 214 "mixing depth" is calculated from the simulated vertical profile of aerosol mass. Is this related to mixed layer height (e.g. height of convective planetary boundary layer)?

If Figure 3 is a representative sample of the simulations it seems as if majority of the plumes are not in the mixed layer but above it, as they do not reach the surface. Again, I would expect the turbulent mixing in convective PBL (or the lack of convective mixing in free troposphere or the residual layer) to have a significant effect on plume dilution and therefore the coagulation rate. Is it so?

Minor comments

Line 69-71 There are some more recent studies which you might want to look up. For instance Akagi et al. (2012), Hennigan et al. (2012), Ortega et al. (2013), Vakkari et al. (2014), Jolleys et al. (2015), Konovalov et al. (2015) and May et al. (2015) come to

my mind.

Line 71 "This SOA condenses onto existing particles causing growth of the aerosol size distribution." Please reconsider this statement as there are observations of new particle formation in biomass burning plumes (Hennigan et al., 2012; Vakkari et al., 2014).

Line 153 "Mixing depth had a range of 150-2500 m" but Table 1 (page 23) gives "Mixing depth" limits as 120 m and 2500 m. Which one is it?

Line 169-170 "The algorithm simulated the size distribution across 15 logarithmically-spaced size bins spanning 3 nm-10 $\mu$m." This leaves quite few bins for the size range of interest. Can coarse size resolution become an issue for the coagulation calculation?

Line 191-192 "We ran 100 SAM-TOMAS simulations at 500 m x 500 m horizontal res-olution (total horizontal extent = 100 km)," but Figure 4 x-axis extends to > 200 km. I assume these are the same data because on lines 244-245 it is stated that "Figure 4 shows the Dpm (panels a and c) and $\sigma$ (panels b and d) as a function of distance for each of the 100 SAM-TOMAS simulations used to train the emulator (Sect. 3.2)." What was the horizontal extent?

Line 382-385 Also for this statement some more recent references could be consid-ered.

Line 824, Figure 3 Please provide a legend for the lines (indicating input and meteoro-logical parameters).

Line 871, Figure 4 There are so many overlying lines that it is getting difficult to read. Please consider if you can clarify it.

References

Akagi, S. K., Craven, J. S., Taylor, J. W., McMeeking, G. R., Yokelson, R. J., Burling, I. R., Urbanski, S. P., Wold, C. E., Seinfeld, J. H., Coe, H., Alvarado, M. J. and Weise,

D. R.: Evolution of trace gases and particles emitted by a chaparral fire in California, Atmos. Chem. Phys., 12(3), 1397–1421, doi:10.5194/acp-12-1397-2012, 2012.

Hennigan, C. J., Westervelt, D. M., Riipinen, I., Engelhart, G. J., Lee, T., Collett, J. L., Pandis, S. N., Adams, P. J. and Robinson, A. L.: New particle formation and growth in biomass burning plumes: An important source of cloud condensation nuclei, Geophys. Res. Lett., 39(9), L09805, doi:10.1029/2012GL050930, 2012.

Jolleys, M. D., Coe, H., McFiggans, G., Taylor, J. W., O'Shea, S. J., Le Breton, M., Bauguitte, S. J.-B., Moller, S., Di Carlo, P., Aruffo, E., Palmer, P. I., Lee, J. D., Percival, C. J. and Gallagher, M. W.: Properties and evolution of biomass burning organic aerosol from Canadian boreal forest fires, Atmos. Chem. Phys., 15(6), 3077–3095, doi:10.5194/acp-15-3077-2015, 2015.

Konovalov, I. B., Beekmann, M., Berezin, E. V., Petetin, H., Mielonen, T., Kuznetsova, I. N. and Andreae, M. O.: The role of semi-volatile organic compounds in the mesoscale evolution of biomass burning aerosol: a modeling case study of the 2010 mega-fire event in Russia, Atmos. Chem. Phys., 15(23), 13269–13297, doi:10.5194/acp-15-13269-2015, 2015.

May, A. A., Lee, T., McMeeking, G. R., Akagi, S., Sullivan, A. P., Urbanski, S., Yokelson, R. J. and Kreidenweis, S. M.: Observations and analysis of organic aerosol evolution in some prescribed fire smoke plumes, Atmos. Chem. Phys., 15(11), 6323–6335, doi:10.5194/acp-15-6323-2015, 2015.

Ortega, A. M., Day, D. A., Cubison, M. J., Brune, W. H., Bon, D., de Gouw, J. A. and Jimenez, J. L.: Secondary organic aerosol formation and primary organic aerosol oxidation from biomass-burning smoke in a flow reactor during FLAME-3, Atmos. Chem. Phys., 13(22), 11551–11571, doi:10.5194/acp-13-11551-2013, 2013.

Pierce, J. R. and Adams, P. J.: Uncertainty in global CCN concentrations from uncertain aerosol nucleation and primary emission rates, Atmos. Chem. Phys., 9(4), 1339–1356,

doi:10.5194/acp-9-1339-2009, 2009.

Vakkari, V., Kerminen, V.-M., Beukes, J. P., Tiitta, P., van Zyl, P. G., Josipovic, M., Venter, A. D., Jaars, K., Worsnop, D. R., Kulmala, M. and Laakso, L.: Rapid changes in biomass burning aerosols by atmospheric oxidation, Geophys. Res. Lett., 41(7), 2644–2651, doi:10.1002/2014GL059396, 2014.

Yokelson, R. J., Crounse, J. D., DeCarlo, P. F., Karl, T., Urbanski, S., Atlas, E., Campos, T., Shinozuka, Y., Kapustin, V., Clarke, A. D., Weinheimer, A., Knapp, D. J., Montzka, D. D., Holloway, J., Weibring, P., Flocke, F., Zheng, W., Toohey, D., Wennberg, P. O., Wiedinmyer, C., Mauldin, L., Fried, A., Richter, D., Walega, J., Jimenez, J. L., Adachi, K., Buseck, P. R., Hall, S. R. and Shetter, R.: Emissions from biomass burning in the Yucatan, Atmos. Chem. Phys., 9(15), 5785–5812, doi:10.5194/acp-9-5785-2009, 2009.

---

## Author Comment (AC1) · 2 Jun 2016

Response to reviewer #2: We'd like to thank the reviewer for their critiques. Our responses to individual comments are below. In addition to the proposed changes to the manuscript, we have added a section (Section 3.6, Figures 11-13) in which the parameterizations for Dpm and sigma are tested against real smoke plumes observed at the Mount Bachelor Observatory (contribution by J.R. Laing and D.A. Jaffe).

*This paper investigates the influence of coagulation on the particle number size distribution, notably on the mean diameter (Dm) and geometric standard deviation (Sigma) of a single particle mode, in biomass burning plumes. The work is based on a large number of sophisticated model simulations. The authors investigate how Dm and Sigma evolve with time in biomass burning plumes, and how their evolution is related to several parameters associated with primary particle emissions, fire conditions and atmospheric conditions. The authors compare briefly the influence of coagulation to that caused by organic aerosol formation/loss in a plume. The authors finally parameterize their results to a form that is applicable in large-scale modeling frameworks.*

*The is scientifically sound and original. The text is well organized and easy to read (with a couple of minor exceptions mentioned below). The authors are able to explain very well the numerous results obtained from model simulations. I have only a few minor suggestions for revisions.*

*Scientific comments:*
*I have a hard time of understanding Figure 3, even after reading the text on lines 210-213. I recommend that the authors work a bit more to make their message here clearer to readers.*

In response to this comment and a comment by reviewer 1, we have simplified Figure 3 to show only two representative profiles of SAM-TOMAS simulations: one which shows the aerosol plume mixing through the boundary layer to the ground, and a second which shows the plume still suspended at the emission injection height at the end of the simulated run. These are the two "types" of mixing depths possible (to ground through the PBL and suspended).

The Figure 3 caption was updated to reflect the changed figure:

"Final vertical profiles for two representative SAM-TOMAS simulations after four hours, normalized to individual aerosol load and averaged horizontally across the domain. The black profile shows a simulation where the aerosol fully mixed through the boundary layer to the ground with some aerosol trapped in a stable layer above the boundary layer, while the red profile shows a simulation where the aerosol plume still stable at the emission injection layer."

*Lines 254-255. The authors state that the initial mode mean diameter have little effect on Dm. I do not get this point when looking at Figures 4a and c: if Dm is initially large, it seems to typically lead to higher values of Dm at later plume times compared to cases where Dm is initially small. Could the authors specify what they mean here?*

We mean to emphasize that the variability in Dpm can mainly be attributed to factors outside of initial Dpm. While those simulations with higher initial diameters do climb to higher final diameters than others, the final Dpm variability is not principally driven by starting diameter. As the dominant Dpm variability factors form a discussion in Section 3.3, we have removed this line to alleviate confusion.

*Line 390: Is this correct? Condensation of a non-volatile vapor into a single mode tend to narrow this mode, not widen it, as stated here.*

Your are correct, we wrote the opposite of what we meant. Line 390 is now: "These assumptions are imperfect as irreversible condensation (evaporation) decreases (increases) σ…".

*The authors analyze shortly the influence OA production/loss on their results (section 3.5), and discuss also the potential effects of cloud processing (lines 463-469). This is clearly sufficient for these two processes in this paper. However, the authors do not mention at all new particle formation (NPF) that has been estimated to be a frequent process in biomass burning plumes. NPF might have notable effects on aerosol size distribution, and thereby on both Dm and Sigma, in evolving biomass burning plume. The authors should spend at least a few lines on discussing the relevance of this process in biomass burning plumes and on the potential effects of NPF on their results.*

We have changed the following discussion of SOA pathways in Section 1.1 (Lines 77-79):

"This SOA can condense onto existing particles causing growth of the aerosol size distribution. It can also spur new-particle formation in biomass-burning plumes as has been observed in lab studies (Hennigan et al., 2012) and field campaign analyses (Vakkari et al., 2014)."

We have added the following lines to our methods:

"We do not address new-particle formation in biomass-burning plumes in this work. In plumes where new-particle formation in biomass-burning plumes occurs, our parameterizations will underestimate the number of particles and overestimate the mean diameter of the plume particles."

*Technical issues:*

*Line 265: Figures 5 shows Sigma versus Dm rather than Dm versus Sigma.*

*Line 382: "..OA has been. . ."*

These have been corrected in text.

---

## Author Comment (AC2) · 2 Jun 2016

Response to reviewer #1: We'd like to thank the reviewer for their critiques. Our responses to individual comments are below. In addition to the proposed changes to the manuscript, we have added a section (Section 3.6, Figures 11-13) in which the parameterizations for $D_{pm}$ and sigma are tested against real smoke plumes observed at the Mount Bachelor Observatory (contribution by J.R. Laing and D.A. Jaffe).

*This paper studies the effect of coagulation on particle diameter (Dm) and geometric standard deviation (sigma) in biomass burning plumes using a large-eddy simulation model with an online aerosol microphysical module. The topic is timely and the text is well written; however, there are some issues that need to be clarified before this manuscript can be accepted. In my opinion this paper presents a valuable base case for assessing the role of coagulation and condensation in future biomass burning studies.*

*Major comments*

*I have two major concerns regarding this study: the lack of organic aerosol (OA) chemistry and that while interpreting the results little attention is placed on plume dilution and its effect on coagulation.*

*Running the simulations as "coagulation-only" limits the usability of the results. For instance, new particle formation has been observed in biomass burning plumes (Hennigan et al., 2012; Vakkari et al., 2014), as well as up to a factor of 4 mass increase during the first few hours (Vakkari et al., 2014). Also a recent study by Konovalov et al. (2015) suggests that accounting for OA volatility can improve model performance significantly, although over a much longer time scale than what is considered here. Therefore, without OA evaporation and condensation, can the parameterisations in this paper be a good starting point for global and regional scale models (c.f. line 37)?*

We agree that in order to simulate the mass of organic aerosol (or total $PM_{2.5}$) in the plumes properly, the dynamic production and/or loss of organic aerosol must be better understood. Our intention was not to claim to the contrary of this, and we had included the following discussion on line 389-394 of our text:

"One of the limitations of the coagulation-only parameterizations derived in this paper is that they do not include the effects of potential condensation/evaporation of organic aerosol on the aged biomass-burning size distribution. Both condensational growth and evaporative loss of OA has been observed previously in chamber studies and the field due to OA production or evaporation from dilution/chemistry (Cubison et al., 2011; Hecobian et al., 2011; Hennigan et al., 2011; Grieshop et al., 2009; Ortega et al., 2013; Jolleys et al., 2015; Vakkari et al., 2014)."

We agree that it is important to expand this discussion to discuss simulations where aerosol mass is a primary focus and have added the following to the lines above: "Thus, in order to predict biomass-burning aerosol mass, and thus the aerosol size distribution, we must understand how OA evolves in biomass-burning plumes.":

We also have in the text a discussion that a change in diameter through coagulation is different than a change in diameter due to OA condensation/evaporation: "It should be noted, however, that the Dpm growth attributed to OA condensation is not accompanied by a change in particle number (additional OA mass is distributed among existing particles), whereas a similar increase in Dpm growth by coagulation only would have an accompanying decrease in particle number. Thus, the changes to the aerosol size distribution and climatic influence of a size change due to coagulation and condensation are different."

Our intention was to show that the evolution and variability of the mean/median particle diameter seems to be more dependent on the details of coagulation than condensation/evaporation.  For example, if fresh biomass-burning particles have a median diameter of 50 nm, a doubling of OA mass would increase the diameter by 13 nm at most, which cannot explain the sizes of aged biomass-burning particles in the atmosphere.  We did not intend to take away the importance of SOA that has a large impact on aerosol direct effects and PM2.5/health and a secondary importance for aerosol size and the indirect effect.

Finally, the Pierce group along with Matt Alvarado at AER have been recently funded by NSF to extend this work to include SOA formation into subgrid plume parameterizations of biomass burning. Adding SOA will take significant effort due to its complex nature and variability between plumes, and we did not feel that we had the time to test SOA properly during Kim Sakamoto's Masters thesis, which ultimately dictated the scope of this work. We hope to have an updated scheme that includes SOA within the next several years that draws upon recent advances of biomass-burning SOA measurements from the lab and field.

*On lines 453-455 it is concluded that SOA formation within the plume has a minor effect compared to coagulation. However, this is based on the assumption that SOA formation does not alter coagulation rate or sigma (lines 388-389). Could you elaborate on the conditions when these assumptions hold? For instance Pierce and Adams (2009) showed that secondary aerosol formation rate is one of the key parameters affecting how large fraction of small particles can grow up to CCN-sizes in new particle formation.*

We have added the following sentence to our text at line 393: "For a 25% growth in diameter from SOA, which may be expected from for a factor of 2 increase in OA mass with a small change in sigma, we expect coagulation rates to stay within about 10% (Seinfeld and Pandis, 2006)."

It is correct that SOA can grow new particles to CCN sizes.  However, this requires much more than a factor-of-2 increase in OA mass as might be expected in biomass-burning plumes.  For example, a factor-of-8 increase in OA mass due to SOA production is required to grow a 40 nm particle to 80 nm. It is certainly plausible that this level of SOA production occurs in some biomass-burning plumes over longer timescales than recent studies; however, this is beyond the scope of this work.

*My second major concern is related to the effect of plume dilution on the coagulation rate. Coagulation depends strongly on aerosol particle number concentration (as stated on lines 145-146). However, the observed changes in Dm and sigma are not discussed in terms of concentration, but only with respect to the input parameters and a rather arbitrary dM/dx (aerosol mass in an infinitesimally thin slice of air perpendicular to wind direction). The effect of dilution on coagulation is mentioned only briefly (e.g. lines 324-326), though Figure 4 shows that in most simulations the Dm and sigma change rapidly near emission, but very slowly later on. Is this decrease in the rate of change in Dm and sigma due to plume dilution and subsequent slowing of coagulation rate?*

The aerosol concentration in-plume has two competing reducers: coagulation and plume dilution (both of which we'd expect to contribute to the rate changes in Figure 4). We expect the coagulation rate to slow as coagulation proceeds and number concentration drops (mean particle diameter increases). Plume dilution has a similar effect of reducing number concentration (and slowing coagulation) without a corresponding increase in particle diameter. Which of these factors dominates is primarily dependent on the size of the plume and the stability of the atmosphere. Large-diameter plumes are more resistant to plume dilution than small-diameter plumes. While we do not track dilution rate as a parameter, a lot of the associated variability is captured by the "Fire Area" parameter (large-area fires have larger-diameter plumes). We do not include dilution as an explicit input parameter in our analysis (even though plume dilution does occur in the LES simulations) because dilution is not an inherent property that would be provided by a coarse-grid aerosol model or an emissions inventory (unlike all of the other parameters that we studied), and thus we think it would be generally useless to have "dilution" as an input to a parameterization that will be used in coarse-grid models. We have added the following discussion of the importance of dilution in Line 81-82:

"The coagulational rate is therefore also affected by the rate of plume dilution (through a reduction in N), itself a function of plume size and meteorological conditions."

While dM/dx and dM/dxdz are somewhat arbitrary, we wanted a parameter which i) captured a large chunk of both Dpm and sigma variability and ii) was dependent on initial conditions only. These parameters only incorporate total aerosol mass loading, and plume vertical and with-wind extent. When dM/dv (initial concentration) was used in the simple parameterization, it was a slightly worse predictor of Dpm and sigma than dM/dxdz. The is because dM/dxdz is the product of dM/dv and the initial plume width; since wider plumes are less susceptible to dilution than narrower plumes, dM/dxdz captures this plume-width effect while dM/dv does not. This is discussed in Lines 393-396:

"dM/dV was also tested as a parameter within these simplified parameterization, but did not yield better agreements for either $D_{pm}$ or σ than dM/dxdz despite incorporating an additional plume parameter (initial plume y-extent). This is because dM/dxdz is the product of dM/dV and the initial plume width; since wider plumes are less susceptible to dilution than narrower plumes, dM/dxdz captures this plume-width effect while dM/dV does not."

*How does Figure 4 look like if you colour it with concentration instead of dM/dx, or plot*

*Dm and sigma against concentration? The dM/dx takes into account only dilution along the wind direction, not dilution due to vertical or cross-wind mixing.*

Figure 4 does not look qualitatively different if either dM/dxdz, dM/dx, or dM/dv (concentration) are used (see attached). We chose dM/dx as it was the simplest parameter in which the Dm plot shows a qualitative trend. We have updated the figure to dM/dxdz instead of dM/dx to illustrate the stability of this trend when accounting for dilution in the vertical. Initial concentration (dM/dV) was not used for consistency with Figures 9-10 where dM/dV is not an improvement over dM/dxdz in the simple parameterization (for the reasons stated above) despite requiring more fire information.

*Can you identify a range (time and space), where coagulation can cause significant changes in the size distribution and after which the plumes become so diluted that coagulation slows down? How would this turning point depend on the initial concentration (emissions) and the meteorological conditions (turbulent mixing) during transport?*

Coagulation rates are always slowing down in plumes (proportional to the square of number concentration), so we would need define that it slows down to a subjectively chosen rate. Whether or not a chosen cutoff rate is appropriate depends on the timescale that one cares about, which depends on the modeling application.  Since this answer also depends on all of the input factors studied here, the answer is not straightforward and is out of the scope of this paper.

*The background aerosol is assumed to be negligible compared to the plume and is set to zero (lines 176-179). However, in ambient air measurements this assumption cannot be made – see e.g. Yokelson et al. (2009). Have you verified that your plumes are so concentrated even after 200km transport that this assumption still holds? When will coagulation rate with the background aerosol become similar to coagulation within the biomass burning mode?*

At the point where the plumes are diluted to ambient-air concentrations, subgrid processing is no longer different from standard grid-size processing and model schemes are sufficient (no parameterization is needed). We have added an explanation of this at Line 178:

"In cases where the plume dilutes to similar concentrations to the ambient background, subgrid-plume coagulation schemes are no longer necessary, and grid-resolved coagulation will properly account for coagulation."

*How is turbulent mixing handled in the simulations? Table 1 (page 23) lists "Mixing depth of aerosol layer" as an input parameter, yet on line 214 "mixing depth" is calculated from the simulated vertical profile of aerosol mass. Is this related to mixed layer height (e.g. height of convective planetary boundary layer)?*

SAM calculates turbulence explicitly in the LES using thermodynamic profiles and surface heat/momentum fluxes. There is no assigned PBL in SAM, though it can be calculated from the

resulting profiles. We have used "mixing depth" and "injection height" to refer to the aerosol layer and *not* the convective planetary boundary layer in general (we do not explicitly calculate it anywhere).

*Injection height* is the height at which the biomass-burning aerosol is emitted into the SAM-LES gridboxes after initialization, after which it is subject to turbulent mixing within the model. Depending on meteorological factors, this initialized aerosol layer can remain stable above the PBL or be mixed down through the PBL to the ground (see revised Figure 3). *Mixing depth* is the term used to describe the height extent of the aerosol layer after it is mixed in the LES (at a given point in the LES simulation) to the final time/distance of the simulation, which may vary significantly from the injection height and/or PBL height.

We have changed Line 153 to emphasize our definition of mixing depth:

"...and aerosol mixing depth (hereafter referred to as *mixing depth;* the vertical extent of the aerosol plume)".

Table 1 has been changed to "Mixing Depth" only for consistency with this definition.

*If Figure 3 is a representative sample of the simulations it seems as if majority of the plumes are not in the mixed layer but above it, as they do not reach the surface. Again, I would expect the turbulent mixing in convective PBL (or the lack of convective mixing in free troposphere or the residual layer) to have a significant effect on plume dilution and therefore the coagulation rate. Is it so?*

*Line 824, Figure 3 Please provide a legend for the lines (indicating input and meteorological parameters).*

We chose our 6 profiles at random for the Figure 3.  For 5 of the 6 these cases, the initial height at which we put the plume in the SAM-TOMAS simulation was not in the mixing boundary layer in the model (which is calculated independently by SAM-TOMAS using the meteorological and surface conditions that were randomly chosen for each case).  Some simulations were at night where the mixing layer was very small <100m and other cases the injection heights were higher than the daytime mixing layer.  We do not expect this to have a great impact on our results as the results should be most sensitive to the depth of the layer the plume mixes to and not whether the plume touches the ground or not.

SAM-TOMAS incorporates a multitude of meteorological parameters (e.g. atmospheric stability) for each simulation in addition to the 7-inputs we varied. Instead of citing specific simulations, we simplified Figure 3 to two representative SAM-TOMAS vertical profiles - one showing the case where the aerosol plume mixes down to the ground through the mixing layer, and one where the plume remains suspended at the injection height throughout the simulation.

The following caption now accompanies Figure 3:

"Final vertical profiles for two representative SAM-TOMAS simulations after four hours, normalized to individual aerosol load and averaged horizontally across the domain. The black profile shows a simulation where the aerosol fully mixed through the boundary layer to the ground, while the red profile shows a simulation where the aerosol plume still stable at the emission injection layer."

Minor comments:
*Line 69-71 There are some more recent studies which you might want to look up. For instance Akagi et al. (2012), Hennigan et al. (2012), Ortega et al. (2013), Vakkari et al. (2014), Jolleys et al. (2015), Konovalov et al. (2015) and May et al. (2015) come tomy mind.*

These references have been added to the appropriate lines with the exception of Ortega et al. (2013),which was already cited at these lines. Konovalov et al. (2015) was added to Section 3.5:

"Konovalov et al. (2015) has emphasized the importance of OA simulation in modeling in long-range (>1000 km) plume evolution."

*Line 382-385 Also for this statement some more recent references could be considered.*

More recent studies have been appended.

*Line 71 "This SOA condenses onto existing particles causing growth of the aerosol size distribution." Please reconsider this statement as there are observations of new particle formation in biomass burning plumes (Hennigan et al., 2012; Vakkari et al., 2014).*

We have added the following discussion of new-particle formation in biomass-burning plumes in lines 71-74:
"This SOA can condense onto existing particles causing growth of the aerosol size distribution. It can also spur new-particle formation in biomass-burning plumes as has been observed in lab studies (Hennigan et al., 2012) and field campaign analyses (Vakkari et al., 2014)."

And in Lines 233-235:

"We do not address new-particle formation in biomass-burning plumes in this work. In plumes where new-particle formation in biomass-burning plumes occurs, our parameterizations will underestimate the number of particles and overestimate the mean diameter of the plume particles."

*Line 153 "Mixing depth had a range of 150-2500 m" but Table 1 (page 23) gives "Mixing depth" limits as 120 m and 2500 m. Which one is it?*

The mixing-depth limit is 150m, and we have updated Table 1.

*Line 169-170 "The algorithm simulated the size distribution across 15 logarithmically-spaced size bins spanning 3 nm-10 µm." This leaves quite few bins for the size range of interest. Can coarse size resolution become an issue for the coagulation calculation?*

There are 10 bins between 10 nm and 1 µm capture the area of biomass-burning aerosol emission and growth. There are only two bins above 1 µm for coarse aerosol. As we track two independent moments (number and mass) within each bin, the model fidelity is much higher than single-moment schemes with similar resolution. The TOMAS microphysics algorithm has been evaluated in Lee and Adams, (2012) and it generally captures the processes similarly to higher-resolution versions of the model. We now clarify the bin structure: "The algorithm simulated the size distribution across 135 logarithmically -spaced size bins spanning 3 nm-1 µm with 2 additional bins spanning 1-10 µm."

The coarse size resolution can have an impact on the sigma calculation of the mode near smaller modal sizes (~1.32) where the distribution is less than a single bin-width. This is discussed in lines 263-265.

*Line 191-192 "We ran 100 SAM-TOMAS simulations at 500 m x 500 m horizontal resolution (total horizontal extent = 100 km)," but Figure 4 x-axis extends to > 200 km. I assume these are the same data because on lines 244-245 it is stated that "Figure 4 shows the Dpm (panels a and c) and σ (panels b and d) as a function of distance for each of the 100 SAM-TOMAS simulations used to train the emulator (Sect. 3.2)." What was the horizontal extent?*

By "total horizontal extent" we were referring to the cross-wind (y-direction) extent, not the with-wind (x-direction) extent as plotted in Figure 4. Lines 190-196 have been clarified to:

"We ran 100 SAM-TOMAS simulations at 500 m x 500 m horizontal resolution (total cross-wind (y-direction) horizontal extent = 100 km), and constant 40 m vertical resolution (total vertical extent = 4 km)... The output from each SAM-TOMAS simulation was recorded at four different times (400 total time slices across 100 simulations) as the plume progressed along the with-wind (x-direction) axis."

*Line 871, Figure 4 There are so many overlying lines that it is getting difficult to read. Please consider if you can clarify it.*

We like that Figure 4 shows the evolution of diameter and sigma across all of our simulations so that the reader can see what variability is captured by our simulations. We dedicate the rest of the paper to generalizing the results, so we prefer to keep Figure similar (though we have updated to use dM/dxdz rather than dM/dx).